# Predicting User Behaviors with Scene via Dual Sequence Networks

## Abstract

Modeling sequential user behaviors for future action prediction is crucial in improving user's information retrieval experience. Recent studies highlight the importance of incorporating contextual information to enhance prediction performance. One crucial and typical contextual information is the scene feature which we define it as sub-interfaces within an app, created by designers to provide specific functionalities, such as "text2product search" and "live" in e-commence apps. Different scenes exhibit distinct functionalities and usage habits, leading to significant distribution gap in user engagement across them. Popular sequential behavior models either ignore the scene feature or merely use it as attribute embeddings, which could lead to substantial information loss or cannot capture the interplay between scene and item in modeling dynamic user interests. In this work, we propose a novel Dual Sequence Prediction network (DSPnet) to effectively capture the interplay between scene and item sequences for future behavior prediction. DSPnet consists of two parallel networks dedicated to predicting scene and item sequences, and a sequence feature enhancement module to capture the interplay. Further, considering the randomness and noise in learning sequence dynamics, we introduce Conditional Contrastive Regularization (CCR) loss to capture the invariance of similar historical sequences. Theoretical analysis suggests that DSPnet can learn the joint relationships between scene and item sequences, and also show better robustness on real-world user behaviors. Extensive experiments are conducted on one public benchmark and two collected industrial datasets. The codes and collected datasets will be made public soon.

## 1 Introduction

Modern online information retrieval services, such as search and recommendation, have brought great changes and convenience for human's daily life. Correspondingly, users' sequential behaviors spread over a variety of apps and websites (Kang & McAuley, 2018; Chen et al., 2021a). Modeling these sequential user behaviors as representations for future behavior prediction has become an important issue in machine learning applications, which greatly improves the downstream services.

Recent advances in modeling sequential user behaviors concentrate on three key areas: *design of the encoding architecture*, *formulation of the training objective* and *utilization of the contextual information*. In *design of the encoding architecture*, early works employ Markov models (Rendle et al., 2010) to capture sequential patterns within historical behavior sequences. However, these models face limitations in their ability to represent complex and higher-order sequential dependencies. Consequently, researchers tend to investigate the more expressive recurrent neural networks (RNNs) (Medsker et al., 2001; Hidasi et al., 2016; Hidasi & Karatzoglou, 2018; Donkers et al., 2017) or self-attention mechanisms (Vaswani et al., 2017; Kang & McAuley, 2018; Sun et al., 2019), to enhance sequential behavior modeling. Subsequently, researchers explored more advanced *formulation of training objective*, beyond the conventional next-item prediction objective. They primarily designed various self-supervised training tasks to extract additional insights from sequences during training (Sun et al., 2019; Yao et al., 2021; Zhou et al., 2020; Fu et al., 2023; Wang et al., 2023a). For instance, Wang et al. (2023a) introduced a context-context contrast which encourages sequences after augmentation to have similar representations by leveraging contrastive learning loss. Additionally, several studies have focused on *utilization of the contextual information* such as item cate-

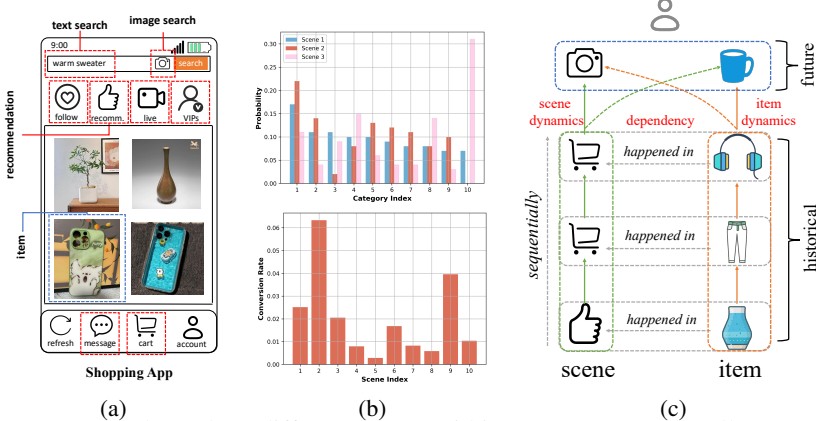

Figure 1: (a) An example to show different scenes within an App. Users usually engage in interactions across different scenes. The red dashed boxes represent distinct scenes, while the blue dashed boxes indicate individual items. (b) An example to show the distribution gap among scenes. The upper one indicates category distribution of different scenes in our e-commence app. The lower one shows users' conversion rate in different scenes. Since the volume of our scenes and categories is quite large, we select several largest scenes and categories for better visualization. (c) shows our idea of performing scene-aware sequential user behavior prediction.

gory (Cai et al., 2021), behavior type (Huang et al., 2018; Chen et al., 2023a) and time intervals (Ye et al., 2020), as the contextual information notably influences user behaviors.

Different from the above contextual feature, one crucial and typical contextual factor influencing user behaviors is the scene feature, which we define it as sub-interfaces created by designers to encapsulate specific themes or functionalities within apps or websites. As shown in Figure 1 (a), different scenes, usually operated by different teams, have different themes and styles. For example, the shopping app encompasses scenes such as "text search", "recommendation" and "live", facilitating functionalities like text-to-product search, product recommendations, and interactive live shopping experiences. *Each of these scenes represents different shopping types, leading to significant distribution gap in user engagement across them.* Figure 1 (b) illustrates the category distribution and user conversion rate across different scenes, unveiling significant disparities in both item content and user engagement among various scenes. The distinct features of a scene play a crucial role in providing conditional information for behavior occurrence. When a user enters a specific scene, it can reflect certain shopping interests and the interests greatly impact the items the user is likely to interact with. *Ignoring the scene feature would result in a large loss of information and introduce data bias in modeling the sequential user trajectory.* Currently, there are limited studies addressing this area because of inaccessible data. *There are remains two challenges to effectively incorporate this scene feature in sequential behavior modeling. One challenge is that how to capture the interplay between scenes and items.* Although we may incorporate the scene feature as an input field of items by following (Tian et al., 2023; Papso, 2023) in user behavior modeling, it could overlook the interplay between scene and item, which hinders their ability of mining scene feature in behavior prediction. As shown in Figure 1 (c), the item and scene simultaneously occur in an interaction behavior, and the sequential scene dynamics and item dynamics have mutual effects in subsequent behavior generation. *The other challenge is that how to better learn sequence dynamics by considering the scene-item misalignment issue within a sequence.* To be more specific, the misalignment issue means a user's interest and intent are initially reflected in one scene but the behavior are incorrectly collected in another scene. For example, in an e-shopping app, a user might see certain products in the "recommendation" interface with the intention to purchase them. However, instead of buying immediately, the user adds these products to the "cart" interface and buys from the "cart" interface later, because of upcoming sales promotions from sellers. The misalignment issue brings obstacles when modeling the interplay between scenes and items.

To this end, we propose a novel Dual Sequence Prediction network (DSPnet) that effectively captures the interplay between sequential scenes and items, while being robust against against scene-item misalignment noise to predict future user behaviors. *DSPnet comprises two parallel networks dedicated to predicting scene and item sequences, together with a sequence feature enhancement module to deliver the mutual effects across both sequences.* In particular, the scene sequence pre-

diction network and item sequence prediction network encode their own dynamics from historical behaviors in sequence level, which avoids the one-to-one scene-item misalignment noise in learning. Meanwhile, the sequence feature enhancement module enables one network's encoding features to be input into the other, allowing both prediction networks to capitalize on their interplay during the sequence learning process. We also demonstrate that the learning approach of DSPnet is theoretically equivalent to maximizing the joint log-likelihood of scenes and items, presenting a good way to model their relationships and inter-dependent sequential dynamics. Moreover, given that sequential user behaviors often exhibit randomness and noise, which can adversely affect the learning of sequence dynamics, we introduce Conditional Contrastive Regularization (CCR) loss to capture the representation invariance of similar historical sequences. Through learned conditional weights, CCR loss can adaptively promote similarity in representations for sequences that undergo augmentation with different forces. We empirically demonstrate that CCR loss highlights the relationships among contrasting samples, enhancing the model's robustness in representation learning for real-world, skewed user behaviors. The contributions of this work are summarized as follows:

- We propose a novel DSPnet method that enhances behavior prediction by capturing the interplay between scenes and items in a sequence. Our theoretical analysis reveals that training DSPnet is equivalent to maximizing the joint log-likelihood of both scene and item sequences, enabling us to effectively model their relationships.

- Further, we introduce Conditional Contrastive Regularization (CCR) to enhance the model's representation learning by capturing the invariance of similar historical sequences. CCR uses learned conditional weights to more effectively promote similarity among those sequences, improving representation robustness in skewed user behaviors.

- We have collected 37-day sequential user behavior data from our e-commence app and constructed two datasets. They contain chronological purchase behaviors on nearly thirty million items, providing a valuable resource to address the research data gap in this field.

- We conduct extensive experiments on three datasets: one public benchmark and two industrial ones. Results on these datasets show the impact of employing scene information in sequential behavior modeling and how our method outperforms state-of-the-art baselines.

## 2 RELATED WORK

**Design of the Encoding Architecture**: Early works investigate Markov chains (Ching & Ng, 2006) to capture sequential dynamics within historical sequential behaviors. However, as the number of past actions increases, the state space grows exponentially, making it challenging to capture higher-order dependencies in real-world applications. Consequently, researchers explore more expressive neural sequence models like Recurrent Neural Networks (RNNs) (Medsker et al., 2001; Hidasi et al., 2016; Hidasi & Karatzoglou, 2018; Donkers et al., 2017), Convolutional Neural Networks (CNNs) (Tang & Wang, 2018), Long Short-Term Memory Networks (LSTM) (Graves & Graves, 2012; Duan et al., 2023) and self-attention (Vaswani et al., 2017; Kang & McAuley, 2018; Sun et al., 2019) models to enhance sequential behavior modeling. For example, SASRec (Kang & McAuley, 2018) and BERT4Rec (Sun et al., 2019) broadened the application of self-attention models to sequential behavior modeling. Some works (Hu et al., 2024; Li et al., 2024b; Zheng et al., 2024; Liao et al., 2024) focus on leveraging large LLMs for sequential recommendation, including aligning sequential RS with LLMs, summarizing user preferences. Others (Ma et al., 2024; Yang et al., 2024; Wang et al., 2024) investigate the application of diffusion models in sequential recommendation, aiming to better capture the evolution of user preferences over time.

**Formulation of the Training Objective**: Several studies concentrate on forecasting item lists over specific time periods or behavioral distributions instead of the next individual items. SUMN (Gu et al., 2021) hypothesizes that future behavioral distributions should align with past distributions. It learns sequence representations by maximizing the Kullback-Leibler divergence between item occurrence distributions from a previous period and those of the future. MSDP (Fu et al., 2023) uses a multi-scale approach to optimize predictions for the next period by considering item lists across different timeframes. Constructing self-supervised learning tasks to facilitate the prediction of sequential user behaviors has also gained considerable attention. CL4SRec (Qiu et al., 2021) explores the contrastive signals derived from augmented historical sequences through contrastive learning. ContraRec (Wang et al., 2023b) achieves state-of-the-art performance by constructing contrastive sequences using random mask and reorder augmentation techniques.

While contrastive learning improves sequential behavior modeling, it often ignores the varying roles of positive and negative samples. Our CCR loss learns conditional weights for these samples, capturing their unique contributions and enhancing the model's robustness in learning sequence dynamics.

**Utilization of the Contextual Information**: In user behavior sequences, there are various contextual factors linked to each action, such as types of user behavior (e.g., clicks, purchases, additions to favorites) (Meng et al., 2020; Ni et al., 2018; Su et al., 2023; Xia et al., 2023; Xuan et al., 2023; Chen et al., 2023a), product category (Cai et al., 2021) and other multiple item attributes (Papso, 2023). DUPN (Ni et al., 2018) incorporates multiple kinds of behavior types to construct multi-task learning for more effective personalization. Xuan et al. (2023) designed the multi-behavior learning module to extract users' personalized information for user-embedding enhancement, and utilize knowledge graph in the knowledge enhancement module to derive more robust knowledge-aware representations for items. MKM-SR (Meng et al., 2020) points out that a user's sequence behaviors could have some micro-behaviors that reflect fine-grained and deep understanding of the user's preference. CoCoRec (Cai et al., 2021) leverages item category to organize a user's own past actions and further employs self-attention to capture in-category transition patterns. Then, these transition patterns are used to find similar users, thereby enhancing collaborative learning. CARCA (Papso, 2023) incorporates both the attributes of interacted items and contextual data of user interactions by employing combined sequences as input for multi-head attention blocks. Some other works employ different scene definitions from ours. Chen et al. (2021b) explored adaptive sequential recommendation systems (RS) across different domains. Wang et al. (2021) defined the scene as a collection of predefined item categories. Wan et al. (2024) investigated the usage of large language models (LLMs) for real-time sequential RS. Li et al. (2024a) defined scenes as 200 predefined topics, such as "weekend spring outing", "afternoon tea" and "KFC crazy Thursday".

It is notable that above works studied different kinds of contextual information from our defined scene feature. Integrating the defined scene features into sequential behavior modeling constitutes an important and new problem setting derived from practical industrial business. When considering techniques, although we may leverage the scene as an additional attribute of items by following (Tian et al., 2023; Papso, 2023), it would ignore the mutual effects between scenes and items and lead to insufficient learning. DSPnet is designed to effectively capture the interplay for behavior prediction.

## 3 DUAL SEQUENCE PREDICTION NETWORK

### 3.1 OVERVIEW

In sequential user behavior prediction task, we aim to predict the user's future behaviors based on historical behaviors. Given a historical behavior sequence $\mathcal{T}$ from user $\boldsymbol{u}$, it is defined as:

$$\mathcal{T} = \{(\boldsymbol{v}_1, \boldsymbol{s}_1), (\boldsymbol{v}_2, \boldsymbol{s}_2), ..., (\boldsymbol{v}_j, \boldsymbol{s}_j), ..., (\boldsymbol{v}_{|\mathcal{T}|}, \boldsymbol{s}_{|\mathcal{T}|})\}, \tag{1}$$

where $\boldsymbol{v}_j \in \boldsymbol{V} = \{\boldsymbol{v}_1, ..., \boldsymbol{v}_{N_v}\}$ and $\boldsymbol{s}_j \in \boldsymbol{S} = \{\boldsymbol{s}_1, ..., \boldsymbol{s}_{N_s}\}$ denote one interacted item and the corresponding scene, respectively. $\boldsymbol{V}$ denotes the whole item set with size $N_v$ and $\boldsymbol{S}$ denotes the whole scene set with size $N_s$. $|\mathcal{T}|$ is the number of historical interactions. The historical behaviors actually contains two coupling sequences, *i.e.* the item sequence $\mathcal{V} = \{\boldsymbol{v}_1, \boldsymbol{v}_2, ..., \boldsymbol{v}_{|\mathcal{T}|}\}$ and the scene sequence $\mathcal{S} = \{\boldsymbol{s}_1, \boldsymbol{s}_2, ..., \boldsymbol{s}_{|\mathcal{T}|}\}$. Based on these two historical sequences, we learn the sequential user representation $\boldsymbol{z_u}$ and predict future behaviors. To better study the issue, we only consider one-type behavior which means the sequence includes only one-type behavior, *e.g.* "buy".

DSPnet consists of two main and original components: dual sequence learning and conditional contrastive regularization (CCR) loss. Both two components are designed to tackle important challenges. The first one is proposed to effectively encode sequential dynamics against scene-item misalignment noise and deliver these dynamics to both scene and item sides for predicting behaviors. The second one aims to learn representation invariance and dynamically enhance the similarity between representations of similar sequences, thereby improving the model's robustness against random, noisy, and skewed user behaviors. The model architecture is shown in Figure 2, details are given in the following parts.

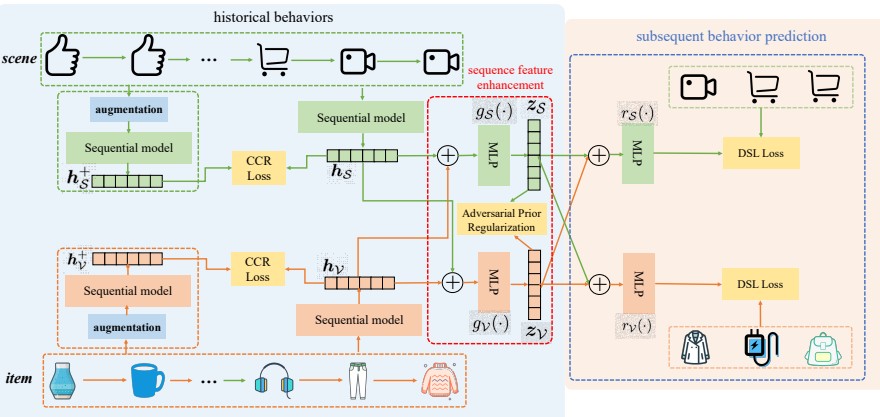

Figure 2: The architecture of DSPnet. Its dual sequence learning models the interplay between scene and item sequences while capturing the sequence dynamics against sscene-item misalignment issues. CCR loss learns representation invariance with different forces on different samples.

## 3.2 DUAL SEQUENCE LEARNING

**Dual Sequence Dynamics**: Online user behaviors occur under different contexts chronologically, reflecting users' dynamic interests over time. The defined scene feature largely influences how users interact with items, while interactions taken on particular items can subsequently influence users' decisions in following scenes. Therefore, the sequential dynamics of both items and scenes play a vital role in predicting subsequent user behaviors. Besides, considering the scene-item misalignment issue in data caused by many factors such as feedback delay and seller promotions, we encode the dynamics in sequence-level correspondence, rather than maintaining strict one-to-one scene-item correspondence. Given the historical scene sequence $\mathcal{S}$ and item sequence $\mathcal{V}$ of user $\boldsymbol{u}$, we can employ some sequential models to capture the sequence dynamics. Let $f_{\mathcal{S}}$ and $f_{\mathcal{V}}$ be the sequential models of historical scenes and items respectively, we have:

$$\boldsymbol{h}_{\mathcal{S}} = f_{\mathcal{S}}(\mathcal{S}), \ \boldsymbol{h}_{\mathcal{V}} = f_{\mathcal{V}}(\mathcal{V}), \tag{2}$$

where $\boldsymbol{h}_{\mathcal{S}}$ and $\boldsymbol{h}_{\mathcal{V}}$ mean the encoded latent representations of $\mathcal{S}$ and $\mathcal{V}$, respectively. The choice of sequential model is flexible (such as RNN, LSTM and transformer) and we employ the powerful transformer model following recent works (Sun et al., 2019; Fu et al., 2023; Wang et al., 2023a).

**Sequence Feature Enhancement**: As we previously discussed, both the item and the scene mutually influence subsequent behavior generation. This means that when predicting future interacted items, we cannot solely depend on information from historical item interactions. Similarly, if we predict the subsequent interacted scenes, we cannot only employ the information from past scene interactions. Either the item sequence or the scene sequence serves as the enhanced information for the other to predict future behaviors. Specifically, denoting $\boldsymbol{z}_{\mathcal{S}}$ and $\boldsymbol{z}_{\mathcal{V}}$ as the feature enhanced representation of scene sequence and item sequence, we have:

$$\boldsymbol{z}_{\mathcal{S}} = g_{\mathcal{S}}(\boldsymbol{h}_{\mathcal{S}} \oplus \boldsymbol{h}_{\mathcal{V}}), \ \boldsymbol{z}_{\mathcal{V}} = g_{\mathcal{V}}(\boldsymbol{h}_{\mathcal{S}} \oplus \boldsymbol{h}_{\mathcal{V}}), \tag{3}$$

where $g_{\mathcal{S}}(\cdot)$ and $g_{\mathcal{V}}(\cdot)$ denote the fusion MLP layers. Both $\boldsymbol{z}_{\mathcal{S}}$ and $\boldsymbol{z}_{\mathcal{V}}$ can be considered as the user representation $\boldsymbol{z}_{\boldsymbol{u}}$ that encodes mutual effects and dynamics from historical behaviors. *We maintain these two enhanced representations here to provide diverse aspects of the user interests and better facilitate the subsequent item and scene prediction tasks.*

**Adversarial Prior Regularization**: Since user behaviors usually face severe data sparsity problem and user representations may overfit to some samples, we impose prior regularization on learned user representations. Specifically, we employ adversarial learning (Makhzani et al., 2016a) to ensure a discriminator cannot discriminate the prior distribution and user presentations. This approach is more advantageous than Kullback-Leibler (KL) divergence regularization.

Let $D_{\mathcal{S}}$ and $D_{\mathcal{V}}$ be the discriminator of $\boldsymbol{z}_{\mathcal{S}}$ and $\boldsymbol{z}_{\mathcal{V}}$ respectively, the adversarial learning based prior regularization is written as:

$$\min_{g_{\mathcal{S}}, g_{\mathcal{V}}, f_{\mathcal{S}}, f_{\mathcal{V}}} \max_{D_{\mathcal{S}}, D_{\mathcal{V}}} \mathcal{L}_{\mathrm{APR}} = \mathbb{E}_{\boldsymbol{z}_{\mathcal{S}} \sim p(\boldsymbol{z}_{\mathcal{S}})}[\log D_{\mathcal{S}}(\boldsymbol{z}_{\mathcal{S}})] + \mathbb{E}_{\boldsymbol{z}_{\mathcal{S}} \sim g_{\mathcal{S}(\cdot)}}[\log(1 - D_{\mathcal{S}}(\boldsymbol{z}_{\mathcal{S}}))]$$

$$+ \mathbb{E}_{\boldsymbol{z}_{\mathcal{V}} \sim p(\boldsymbol{z}_{\mathcal{V}})}[\log D_{\mathcal{V}}(\boldsymbol{z}_{\mathcal{V}})] + \mathbb{E}_{\boldsymbol{z}_{\mathcal{V}} \sim g_{\mathcal{V}(\cdot)}}[\log(1 - D_{\mathcal{V}}(\boldsymbol{z}_{\mathcal{V}}))], \tag{4}$$

where $p(\boldsymbol{z}_{\mathcal{S}})$ and $p(\boldsymbol{z}_{\mathcal{V}})$ are the prior distribution. DSPnet includes a behavior prediction task that matches true behavior distributions, largely preventing mode collapse issues in adversarial learning.

**Subsequent Behavior Prediction**: When conducting subsequent behavior prediction, we incorporate both enhanced representations $\boldsymbol{z}_{\mathcal{S}}$ and $\boldsymbol{z}_{\mathcal{V}}$ for behavior prediction:

$$\boldsymbol{o}_{\mathcal{S}} = r_{\mathcal{S}}(\boldsymbol{z}_{\mathcal{S}} \oplus \boldsymbol{z}_{\mathcal{V}}), \ \ \boldsymbol{o}_{\mathcal{V}} = r_{\mathcal{V}}(\boldsymbol{z}_{\mathcal{S}} \oplus \boldsymbol{z}_{\mathcal{V}}), \tag{5}$$

where $\boldsymbol{o}_{\mathcal{S}}$ and $\boldsymbol{o}_{\mathcal{V}}$ are outputs of the feature selection functions. Additionally, since the next user behavior may be stochastic and noisy, we choose to predict the subsequent behaviors over a period of time following (Fu et al., 2023). Here, we denote the candidate scene set and candidate item set for prediction as $\boldsymbol{V}^{\text{cand}} \subseteq \boldsymbol{V}$ and $\boldsymbol{S}^{\text{cand}} \subseteq \boldsymbol{S}$, respectively. The ground-truth label of subsequent scene behavior is given by $\boldsymbol{y^s} \in \{0, 1\}^{K^s}$ and the subsequent item behavior is given by $\boldsymbol{y^v} \in \{0, 1\}^{K^v}$, where $K^s$ and $K^v$ are the size of candidate scene or item set. Then the prediction objective functions on future scenes and items are formulated as follows:

$$\mathcal{L}_{\text{DSL}}^{\mathcal{S}} = -\frac{1}{K} \sum_{k=1}^{K} [\boldsymbol{y}_k^{\boldsymbol{s}} \log(\hat{\boldsymbol{y}_k^{\boldsymbol{s}}}) + (1 - \boldsymbol{y}_k^{\boldsymbol{s}}) \log(1 - \hat{\boldsymbol{y}_k^{\boldsymbol{s}}})], \ \ \hat{\boldsymbol{y}_k^{\boldsymbol{s}}} = \sigma(\boldsymbol{o}_{\mathcal{S}} \cdot \boldsymbol{e}_k^{\boldsymbol{s}}), \tag{6a}$$

$$\mathcal{L}_{\text{DSL}}^{\mathcal{V}} = -\frac{1}{K} \sum_{k=1}^{K} [\boldsymbol{y}_k^{\boldsymbol{v}} \log(\hat{\boldsymbol{y}_k^{\boldsymbol{v}}}) + (1 - \boldsymbol{y}_k^{\boldsymbol{v}}) \log(1 - \hat{\boldsymbol{y}_k^{\boldsymbol{v}}})], \ \ \hat{\boldsymbol{y}_k^{\boldsymbol{v}}} = \sigma(\boldsymbol{o}_{\mathcal{V}} \cdot \boldsymbol{e}_k^{\boldsymbol{v}}), \tag{6b}$$

where $\hat{\boldsymbol{y}_k^{\boldsymbol{s}}}$ and $\hat{\boldsymbol{y}_k^{\boldsymbol{v}}}$ indicate the prediction probability of $k$-th scene and item in candidate sets, $\boldsymbol{e}_k^{\boldsymbol{s}}$ and $\boldsymbol{e}_k^{\boldsymbol{v}}$ are the latent representations of $k$-th candidate scene and item, respectively. $\sigma(\cdot)$ is the sigmoid function and $\cdot$ means the inner-product operation.

We also reveal the theoretical analysis of this dual sequence learning mechanism on capturing the interplay between scene and item sequences.

**Lemma 1** *Without specifying the sequential encoder architecture and prediction objective function, minimizing the dual sequence learning scheme is equivalent to maximizing the following evidence lower bound of the joint log-likelihoods of observed item and scene sequential behaviors:*

$$\max_{\theta_1, \theta_2, \phi_1, \phi_2} \mathcal{L}_{ELBO} = \mathbb{E}_{q_{\phi_1}(\boldsymbol{z}_{\mathcal{V}}|\mathcal{V}, \mathcal{S}) q_{\phi_2}(\boldsymbol{z}_{\mathcal{S}}|\mathcal{V}, \mathcal{S})} [\log p_{\theta_1}(\boldsymbol{v}|\boldsymbol{z}_{\mathcal{V}}, \boldsymbol{z}_{\mathcal{S}}) p_{\theta_2}(\boldsymbol{s}|\boldsymbol{z}_{\mathcal{V}}, \boldsymbol{z}_{\mathcal{S}})]$$

$$- D_{KL}[q_{\phi_1}(\boldsymbol{z}_{\mathcal{V}}|\mathcal{V}, \mathcal{S})||p(\boldsymbol{z}_{\mathcal{V}})] - D_{KL}[q_{\phi_2}(\boldsymbol{z}_{\mathcal{S}}|\mathcal{V}, \mathcal{S})||p(\boldsymbol{z}_{\mathcal{S}})], \tag{7}$$

*where $\boldsymbol{v}$ and $\boldsymbol{s}$ denote the observed item and scene, respectively. $\mathcal{V}$ and $\mathcal{S}$ indicate the historical sequential items and scenes before $\boldsymbol{v}$ and $\boldsymbol{s}$, respectively. $\boldsymbol{z}_{\mathcal{V}}$ and $\boldsymbol{z}_{\mathcal{S}}$ are the encoded representations from historical behaviors. $D_{KL}$ is the KL Divergence that imposes prior regularization.*

Detailed derivation is given in A.1. The meaning of Eq. 7 is equivalent to our model design in Figure 2. To be specific, $q_{\phi_1}(\boldsymbol{z}_{\mathcal{V}}|\mathcal{V}, \mathcal{S})$ and $q_{\phi_2}(\boldsymbol{z}_{\mathcal{S}}|\mathcal{V}, \mathcal{S})$ are the posteriors that encode information from $\mathcal{V}, \mathcal{S}$ into $\boldsymbol{z}_{\mathcal{V}}, \boldsymbol{z}_{\mathcal{S}}$. $p_{\theta_1}(\boldsymbol{v}|\boldsymbol{z}_{\mathcal{V}}, \boldsymbol{z}_{\mathcal{S}})$ and $p_{\theta_2}(\boldsymbol{s}|\boldsymbol{z}_{\mathcal{V}}, \boldsymbol{z}_{\mathcal{S}})$ indicate that predicting the future items or scenes both should be dependent on the historical behaviors, like our design in Eq. 5. The last two KL divergence correspond to our adversarial learning based prior regularization in Eq. 4.

*Remark*: Modeling the joint log-likelihood of scene and item sequences is a principled way to capture their interplay and dynamics, which is usually overlooked in (Sun et al., 2019; Fu et al., 2023; Chen et al., 2019; Wang et al., 2023a). Although some works (Tian et al., 2023; Papso, 2023) can incorporate the scene information as an additional attribute embedding for items, they fail to capture such interplay. In this regard, dual sequence learning can empower our DSPnet to learn more comprehensive representations of historical behaviors for improved behavior predictions.

### 3.3 SEQUENTIAL CONTRASTIVE REPRESENTATION LEARNING

Unlike the structured human language, online sequential user behaviors are often random and noisy. In DSPnet, we use sequential contrastive representation learning to capture the dynamics of sequential behaviors. We create augmented sequences from the original data, align their representations with the originals, and ensure that representations of different sequences remain distinct.

**Sequence Augmentation**: The sequence augmentation must not alter the user's intended meaning in input sequence. Drawing insights from recent studies (Sun et al., 2019; Wang et al., 2023a),

we utilize masking and reordering approaches to perform sequence augmentation. The mask augmentation involves randomly masking a percentage of elements from the input sequence. Reorder augmentation consists of two steps: first, we randomly select a size that ranges from 2 up to the length of the sequence. Then, we uniformly choose a continuous subsequence of this size and shuffle its elements, while the elements outside of this subsequence retain their original order. Let $\mathcal{A}(\cdot)$ represent a function that applies augmentation to the original sequence. We can express the augmented historical scene and item sequence as $\mathcal{S}^+ = \mathcal{A}(\mathcal{S})$ and $\mathcal{V}^+ = \mathcal{A}(\mathcal{V})$, respectively. These augmented samples offer valuable signals for learning representation invariance.

**Conditional Contrastive Regularization**: To learn the invariance of behavior sequences, we aim to maximize the similarity between original and augmented sequences while minimizing the similarity to sampled dissimilar sequences. Additionally, we introduce two conditional weights to reflect the differing contributions of augmented and sampled dissimilar sequences in optimization.

Let $\boldsymbol{h}_\mathcal{V}^+$ be the representation of augmented item sequence $\mathcal{V}^+$ and $\boldsymbol{h}_\mathcal{V}^-$ be the representation of sampled dissimilar item sequence, then our contrastive loss with two conditional weights is:

$$\mathcal{L}_{\text{CCR}}^\mathcal{V} = -\mathbb{E}_{\boldsymbol{h}_\mathcal{V}}[\sum_{i=1}^{N_+} \underbrace{\frac{e^{-s(\boldsymbol{h}_\mathcal{V}, \boldsymbol{h}_{\mathcal{V},i}^+)}}{\sum_i^{N_+} e^{-s(\boldsymbol{h}_\mathcal{V}, \boldsymbol{h}_{\mathcal{V},i}^+)}}}_{\text{conditional weights: } \boldsymbol{w}_{\mathcal{V},i}^+} s(\boldsymbol{h}_\mathcal{V}, \boldsymbol{h}_{\mathcal{V},i}^+)] + \mathbb{E}_{\boldsymbol{h}_\mathcal{V}}[\sum_{j=1}^{N_-} \underbrace{\frac{e^{s(\boldsymbol{h}_\mathcal{V}, \boldsymbol{h}_{\mathcal{V},j}^-)}}{\sum_j^{N_-} e^{s(\boldsymbol{h}_\mathcal{V}, \boldsymbol{h}_{\mathcal{V},j}^-)}}}_{\text{conditional weights: } \boldsymbol{w}_{\mathcal{V},j}^-} s(\boldsymbol{h}_\mathcal{V}, \boldsymbol{h}_{\mathcal{V},j}^-)], \quad (8)$$

where $s(\boldsymbol{h}_\mathcal{V}, \boldsymbol{h}_\mathcal{V}^+) = \boldsymbol{h}_\mathcal{V}^T \boldsymbol{h}_\mathcal{V}^+ / \tau^+$ and $s(\boldsymbol{h}_\mathcal{V}, \boldsymbol{h}_\mathcal{V}^-) = \boldsymbol{h}_\mathcal{V}^T \boldsymbol{h}_\mathcal{V}^- / \tau^-$ calculate the similarity between two vectors. $\tau^+$ and $\tau^-$ are two temperature hyper-parameters. $N_+$ and $N_-$ indicate the number of augmented sequences (*i.e.* positive samples) and dissimilar sequences (*i.e.* negative samples). $\boldsymbol{w}_{\mathcal{V},i}^+$ and $\boldsymbol{w}_{\mathcal{V},j}^-$ are the conditional weights which are designed to mine hard samples to perform more effective representation learning. Note that the conditional contrastive loss of scene sequence $\mathcal{L}_{\text{CCR}}^\mathcal{S}$ can be written in similar formula with $(\boldsymbol{h}_\mathcal{S}, \boldsymbol{h}_\mathcal{S}^+, \boldsymbol{h}_\mathcal{S}^-)$ as input.

*Remark*: Given the original sequence, different augmented sequences could have different contributions in optimization. Meanwhile, when sampling negative samples for skewed data distributions, such as the pronounced long-tailed patterns in user behavior data, the relationships among negatives may be largely different from uniform distribution. Therefore, it is vital to optimize the contrastive signals with conditional weights, unlike the uniform weights in conventional contrastive loss.

### 3.4 TRAINING OBJECTIVE FUNCTION

To sum up, we can write the whole training objective function of DSPnet as follows:

$$\mathcal{L}_{\text{DSPnet}} = \mathcal{L}_{\text{DSL}}^\mathcal{V} + \lambda * \mathcal{L}_{\text{DSL}}^\mathcal{S} + \alpha * \mathcal{L}_{\text{APR}} + \beta * \mathcal{L}_{\text{CCR}}, \quad (9)$$

where $\mathcal{L}_{\text{CCR}} = \mathcal{L}_{\text{CCR}}^\mathcal{S} + \mathcal{L}_{\text{CCR}}^\mathcal{V}$. The $\lambda$, $\alpha$ and $\beta$ are hyper-parameters to weight the importance of loss terms. We usually care more on future item prediction in practice, so we take $\mathcal{L}_{\text{DSL}}^\mathcal{V}$ to be the main part and set $\lambda$ on $\mathcal{L}_{\text{DSL}}^\mathcal{S}$ here. To sum up, DSPnet offers an efficient and principled approach for modeling the interplay and dynamics of sequential scene and item behaviors.

## 4 EXPERIMENTS AND ANALYSIS

### 4.1 EXPERIMENT SETUP

**Datasets**: We conduct experiments on three datasets, one of which is a public benchmark, while the other two are collected from our e-commence app. The public dataset, Ourbrain[1], focuses on news recommendation and contains chronological views of user interactions with documents. For this dataset, we utilize the view sequence and feature fields "uuid", "document_id", "timestamp", and "source_id". Here, "uuid" identifies the user, while "timestamp" records when an interaction occurred. The "document_id" serves as the item id, and "source_id", linked to the publisher's website, indicates the scene information. We take behaviors before 1975-10-01[2] as historical behaviors

---

[1] https://www.kaggle.com/competitions/outbrain-click-prediction/overview

[2] Note this time is directly transformed from the "timestamp" feature, without adding the actual time offset.

Table 1: Performance comparison of different methods on next item prediction task. R@$k$ and N@$k$ represent Recall@$k$ and NDCG@$k$, respectively. We use "w/o" to denote DSPnet without a particular part. The best results are bolded and the most competitive public baselines are underlined.

| Dataset | Outbrain | | | | AllScenePay-1m | | | | AllScenePay-10m | | | |
|---|---|---|---|---|---|---|---|---|---|---|---|---|
| Method | R@5 | N@5 | R@10 | N@10 | R@5 | N@5 | R@10 | N@10 | R@5 | N@5 | R@10 | N@10 |
| BERT4Rec | 0.0943 | 0.0676 | 0.1384 | 0.0819 | OOM | OOM | OOM | OOM | OOM | OOM | OOM | OOM |
| MSDP | 0.2703 | 0.2181 | 0.2994 | 0.2275 | 0.0006 | 0.0004 | 0.0010 | 0.0005 | 0.0005 | 0.0003 | 0.0011 | 0.0005 |
| ContraRec | 0.3619 | 0.2468 | 0.4701 | 0.2820 | 0.0753 | 0.0533 | 0.1010 | 0.0616 | 0.1414 | 0.1026 | 0.1925 | 0.1191 |
| SceneCTC | 0.4811 | 0.4068 | 0.5232 | 0.4205 | 0.0735 | 0.0517 | 0.1023 | 0.0610 | 0.1459 | 0.1027 | 0.1974 | 0.1193 |
| SceneContraRec | 0.4979 | 0.4027 | 0.5448 | 0.4182 | 0.0762 | 0.0544 | 0.1045 | 0.0635 | 0.1455 | 0.1028 | 0.1983 | 0.1199 |
| CARCA | 0.5126 | 0.4373 | 0.5430 | 0.4472 | OOM | OOM | OOM | OOM | OOM | OOM | OOM | OOM |
| DSPnet– | 0.5324 | 0.4612 | 0.5604 | 0.4703 | 0.0742 | 0.0527 | 0.1047 | 0.0625 | 0.1443 | 0.1028 | 0.1987 | 0.1201 |
| DSPnet(w/o $\mathcal{L}_{APR}, \mathcal{L}_{CCR}$) | 0.6115 | 0.5292 | 0.6625 | 0.5459 | 0.0843 | 0.0617 | 0.1123 | 0.0707 | 0.1680 | 0.1241 | 0.2206 | 0.1411 |
| DSPnet(w/o $\mathcal{L}_{CCR}$) | 0.6109 | 0.5327 | 0.6674 | 0.5511 | 0.0845 | 0.0616 | 0.1121 | 0.0704 | 0.1710 | 0.1266 | 0.2239 | **0.1437** |
| DSPnet(w/o $\mathcal{L}_{APR}$) | 0.6198 | 0.5388 | 0.6684 | **0.5545** | 0.0870 | 0.0630 | **0.1158** | 0.0723 | 0.1711 | 0.1266 | 0.2229 | 0.1433 |
| DSPnet | **0.6248** | 0.5368 | **0.6717** | 0.5520 | **0.0870** | **0.0632** | 0.1155 | **0.0725** | **0.1712** | **0.1267** | **0.2240** | 0.1436 |
| | (+11.22%) | (+9.95%) | (+12.69%) | (+10.48%) | (+1.07%) | (+0.88%) | (+1.10%) | (+0.90%) | (+2.53%) | (+2.39%) | (+2.57%) | (+3.17%) |

and those following as prediction behaviors. We filtered sequences whose number of historical or future actions is less than 1. The dataset is split as train/val/test set with common 8/1/1 setting.

The two industrial datasets are named as AllScenePay-1m and AllScenePay-10m, which contain 1 million and 10 million user purchase sequences, respectively. The occurrence time of these purchase behaviors ranges from 2024-07-01 to 2024-08-07. We take behaviors from 2024-07-01 to 2024-07-31 as historical behaviors and those those from 2024-08-01 to 2024-08-07 as prediction behaviors. We filtered out sequences with fewer than 3 historical or prediction behaviors. To conduct fast evaluation, we randomly select 10% sequences as the val and test set, and the rest are taken as the train set. **More details including statistics on these datasets are given in Appendix Table 3.**

**Baselines**: We make performance comparison with recent strong and popular methods, including the aspect of *encoding architecture design*, *training objective formulation* and *contextual information utilization*. BERT4Rec (Sun et al., 2019) introduces bidirectional self-attention to sequential behavior modeling. MSDP (Fu et al., 2023) introduces a multi-scale stochastic distribution prediction as the training objective. In ContracRec (Wang et al., 2023a) introduces a context-context contrastive loss to make similar sequences learn similar representations. Further, we introduce SceneCTC and SceneContraRec as the baselines incorporating scene information as embeddings like CARCA (Papso, 2023). SceneCTC employs context-target contrastive loss from (Wang et al., 2023a). SceneContraRec extends the input of ContraRec with scene feature. We also add DSPnet– that replaces our dual sequence encoder with one-to-one correspondence encoding in recent multi-behavior sequential RS to study the effectiveness of our dual sequence encoding scheme.

**Parameter Settings**: The dimension of item embeddings and scene embeddings is set as 256 for all models on Outbrain. Since the number of items is too large on our industrial datasets, we set the dimension of item embeddings and scene embeddings as 16 and 4 on AllScenePay-1m and AllScenePay-10m for all models to save computation memory. We use one GPU for training on Outbrain and the batch size is 32. While 8 GPUs are used on the two industrial datasets and the batch size on each GPU is 32. We use the validation performance as early stop condition and the max training epoch is 100. Hyper-parameters of baselines are set according to their papers or searched on our datasets. In DSPnet, we employ the transformer in (Sun et al., 2019) as our sequential model and the transformer layer is 2. The number of MLP layers in $g_{\mathcal{S}}(\cdot)$ and $g_{\mathcal{V}}(\cdot)$ equals 2. The number of positive samples in CCR is 2, and that of negative samples is dependent on the batch size because we use the popular intra-batch sampling to sample negatives. The temperature parameters are set as $\tau^+ = 1.0$ and $\tau^- = 0.07$ by experience. Meanwhile, since the dataset size of Outbrain is small, we set $r_{\mathcal{S}}(\cdot) = r_{\mathcal{V}}(\cdot)$ as one linear MLP. We set $\lambda = 1.0$, $\alpha = 2 \times 10^{-7}$ and $\beta = 5 \times 10^{-6}$ on Outbrain, while $\lambda = 0.2$, $\alpha = 10^{-9}$ and $\beta = 10^{-7}$ on two industrial datasets[3]. The prior distribution is standard Gaussian distribution. The study of other prior distributions is provided in Appendix D.2.

### 4.2 OVERALL COMPARISON

In this section, we present the performance comparison for both the next behavior prediction task in Table 1 and period behavior prediction task in Table 4 of Appendix D. Given that next behavior can be stochastic while behavior distribution over a time period tends to be more stable, we introduce

---

[3]The loss value of $\mathcal{L}_{DSL}$ is quite small due to the abundance of negative samples compared to the few positive ones in BCE calculation. We set $\alpha$ and $\beta$ to a small scale to ensure they do not dominate $\mathcal{L}_{DSL}$.

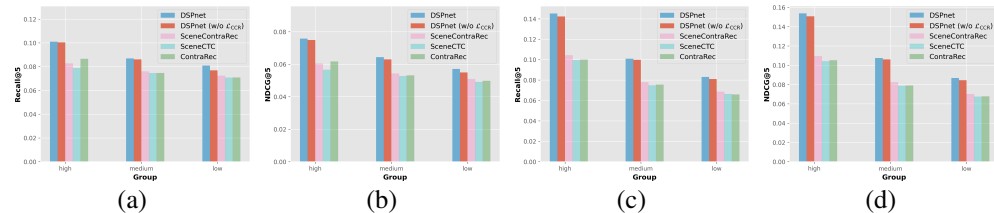

(a)         (b)         (c)         (d)

Figure 3: Performance comparison of different methods on different user groups. (a) and (b) indicate results on next behavior prediction task. (c) and (d) show results on period behavior prediction task.

Table 2: Study of the sequence feature enhancement module.

| Dataset | Ourbrain | | | | AllScenePay-1m | | | | AllScenePay-10m | | | |
|---|---|---|---|---|---|---|---|---|---|---|---|---|
| Model | R@5 | N@5 | R@10 | N@10 | R@5 | N@5 | R@10 | N@10 | R@5 | N@5 | R@10 | N@10 |
| w/o concat | 0.3661 | 0.2997 | 0.4028 | 0.3118 | 0.0701 | 0.0488 | 0.0994 | 0.0582 | 0.1412 | 0.1000 | 0.1953 | 0.1175 |
| w/o MLP | 0.5400 | 0.4719 | 0.5704 | 0.4819 | **0.0899** | 0.0627 | **0.1215** | **0.0729** | 0.1645 | 0.1201 | 0.2184 | 0.1375 |
| DSPnet(MLP_layers=1) | 0.5633 | 0.4795 | 0.6211 | 0.4984 | 0.0868 | 0.0634 | 0.1138 | 0.0720 | 0.1676 | 0.1241 | 0.2194 | 0.1409 |
| DSPnet(MLP_layers=3) | 0.6175 | 0.5360 | 0.6644 | 0.5513 | 0.0857 | 0.0623 | 0.1165 | 0.0723 | 0.1621 | 0.1207 | 0.2111 | 0.1365 |
| DSPnet(MLP_layers=2) | **0.6248** | **0.5368** | **0.6717** | **0.5520** | 0.0870 | **0.0632** | 0.1155 | 0.0725 | **0.1712** | **0.1267** | **0.2240** | **0.1436** |

the period behavior prediction, which focuses on forecasting user behaviors within a time period. We also investigate the impact of model components by removing them.

From the tables, we observe that: 1) combining the scene information can obviously promote the modeling ability of sequential behaviors. By incorporating this information, SceneContraRec improves its Recall@5 score from 0.3619 to 0.4979 on Outbrain. 2) When considering the technique of combining scene information, DSPnet demonstrates a clear advantage over popular methods that only use scene information as attribute embeddings. For instance, DSPnet outperforms SceneContraRec by achieving a 11.22% increase in Recall@5 on Outbrain. The proposed dual sequence learning facilitates the model to capture inter-dependent dynamics between two sequences. Meanwhile, CCR loss enables the model to better learn representation invariance of historical sequences. Note CARCA contains a complex cross attention module and Bert4Rec involves the Cloze task that outputs large memory tensors for loss calculation. They have OOM issue on our large-scale datasets.

When removing $\mathcal{L}_{\text{APR}}$ and $\mathcal{L}_{\text{CCR}}$, we only have the vanilla dual sequence learning $\mathcal{L}_{\text{DSL}}^{\mathcal{V}}$ and $\mathcal{L}_{\text{DSL}}^{\mathcal{S}}$ in working. In this case, DSPnet(w/o $\mathcal{L}_{\text{APR}}$, $\mathcal{L}_{\text{CCR}}$) can still achieve better performance than baselines, emphasizing the effectiveness of our dual sequence learning approach. Additionally, either removing $\mathcal{L}_{\text{APR}}$ or $\mathcal{L}_{\text{CCR}}$ would deteriorate the model performance. $\mathcal{L}_{\text{APR}}$ incorporates prior knowledge into the learned representations, while $\mathcal{L}_{\text{CCR}}$ facilitates the learning of representation invariance, with both contributing to improved behavior prediction. Besides, comparing the performance between DSPnet– and DSPnet, we clearly see our dual sequence scheme does better in behavior prediction. We attribute this to two key factors. First, by incorporating a sequence feature enhancement module, our dual sequence encoder explicitly captures the complex interactions between scene and item. Second, our dual sequence-level encoder is robust against scene-item misalignment errors.

### 4.3 PERFORMANCE ON DIFFERENT USER GROUPS

As the length of user sequences usually follows a severe long-tailed distribution, we conduct an experiment to study the model's generalization ability on different parts of the distribution. We split test user sequences into three groups (*i.e.* "high", "medium", "low") according to their sequence lengths. The comparison results are given in Figure 3.

From this figure, we see DSPnet has consistent improvements on different groups over other baselines. This demonstrates the superior generalization capability of our method. Besides, DSPnet shows a more significant performance gap over DSPnet (w/o $\mathcal{L}_{\text{CCR}}$) for the "low" group than for the "medium" group. As discussed in Section 3.3, CCR loss provides the advantage of considering relationships inner positives or negatives, which is quite important for skewed data distributions.

### 4.4 STUDY OF SEQUENCE FEATURE ENHANCEMENT

In our dual sequence learning, the sequence feature enhancement is an important module to capture the interplay between two historical sequences. We thus conduct an experiment to explore how this component influence the model performance. The results are summarized in Table 2.

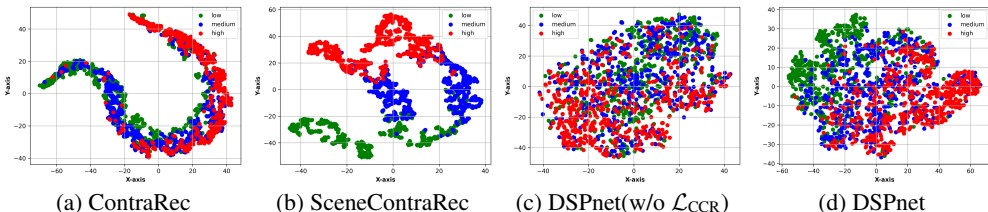

| (a) ContraRec | (b) SceneContraRec | (c) DSPnet(w/o $\mathcal{L}_{CCR}$) | (d) DSPnet |

Figure 4: The t-SNE visualization of learned user representations on AllScenePay-1m dataset.

From this table, we can conclude that: 1) The removal of feature enhancement module (denoted as "w/o concat") leads to a significant decrease in model performance, underscoring the crucial role of sequence feature enhancement module in capturing the interplay and dynamics for predicting future behaviors. 2) When the MLP layers are excluded (as indicated by "w/o MLP"), the model relies solely on concatenation operation to integrate information. This limitation results in poorer performance compared to the variants that include MLP layers, as the "w/o MLP" variant lacks capacity and flexibility to generate fused user representations. 3) Different the number of MLP layers lead to different model performances. With proper MLP layers, we can enhance the model's capability, allowing for better interplay modeling.

### 4.5 Visualization of User Representations

In sequential behavior modeling, user representations are usually encoded from historical behaviors and largely influence the performance of final behavior prediction. We here investigate whether the learned user representations are better than baselines. Specifically, we obtain user representations of test set and split them into three groups (*i.e.* "high", "medium", "low") based on their number of historical interactions. Then, for each group, we randomly sample 500 user representations for t-SNE visualization. The results of different methods are given in Figure 4.

From this figure, we summarize that: 1) User representations generated by DSPnet achieve significant improvements compared to baselines. In ContraRec, representations from different groups are intertwined, leading to less discrimination. SceneContraRec manages to classify representations well, but they tend to converge into a small region, which can be detrimental for personalization in subsequent recommendation or retrieval tasks. Additionally, the representation distances in SceneContraRec do not align with the expectation that distance between "high" and "low" groups should be greater than that between "medium" and "low" groups. In contrast, DSPnet's representations are distinctly differentiable, do not collapse into a small subspace, and exhibit clear distance interpretability. 2) When comparing panels (c) and (d), it is evident that the inclusion of CCR loss facilitates the learning of more compact representations, particularly within the "low" group. CCR loss promotes representation invariance and generalization abilities on skewed user behaviors.

### 5 Conclusion and Future Work

Learning to represent sequential user behaviors for predicting future actions is a crucial topic in machine learning applications. In this study, we propose a novel framework called DSPnet that effectively captures the interplay between historical scene and item sequences, enabling a better modeling of dynamic user interests for future behavior prediction. Additionally, recognizing the randomness and noise inherent in user behaviors, we introduce CCR loss to enhance representation invariance, thereby improving the learning of dynamic interests. Through both theoretical analysis and empirical evaluation, we demonstrate that DSPnet does better at modeling the interplay between sequences and exhibits superior performance in skewed data scenarios.

Although DSPnet has achieved remarkable performance of user behavior prediction, it still has certain limitations. For instance, the current DSPnet only incorporates scene and item sequences, leaving potential to improve performance by integrating additional feature information from these sequences. Additionally, DSPnet currently establishes the interplay between historical scene and item sequences only after processing the last token's representation. An alternative, but more computationally consuming approach, would be to model these interplays at the level of each token's representation. We plan to study these issues in our future research.

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

# A   DERIVATION OF THE THEOREMS

## A.1   DERIVATION OF THE JOINT LOG-LIKELIHOOD

Let $\boldsymbol{v}$ and $\boldsymbol{s}$ denote the observed interacted item and scene of user $\boldsymbol{u}$, respectively. Then, the joint log-likelihood is composed of a sum over the likelihoods of individual data points $\sum_{\boldsymbol{u}}[\log p_\theta(\boldsymbol{v}, \boldsymbol{s})]$, where $p_\theta(\boldsymbol{v}, \boldsymbol{s})$ is the probability density function. Given the observed item-scene behaviors $(\boldsymbol{v}, \boldsymbol{s})$, we denote $\mathcal{V}$ and $\mathcal{S}$ as the historically interacted items and scenes sequentially before $\boldsymbol{v}$ and $\boldsymbol{s}$, respectively. The corresponding encoded latent representations of $\mathcal{V}$ and $\mathcal{S}$ are denoted as $\boldsymbol{z}_\mathcal{V}$ and $\boldsymbol{z}_\mathcal{S}$, respectively. Then, drawing from the idea of maximizing the marginal log-likelihood in Variational Autoencoders (VAEs) (Kingma & Welling, 2013), $\log p_\theta(\boldsymbol{v}, \boldsymbol{s})$ can be written as:

$$\log p_\theta(\boldsymbol{v}, \boldsymbol{s}) = D_{KL}[q_\phi(\boldsymbol{z}_\mathcal{V}, \boldsymbol{z}_\mathcal{S}|\mathcal{V}, \mathcal{S})||p(\boldsymbol{z}_\mathcal{V}, \boldsymbol{z}_\mathcal{S}|\mathcal{V}, \mathcal{S})] + \mathcal{L}_{\text{ELBO}}, \tag{10}$$

where the first term denotes KL divergence between parameterized posterior $q_\phi(\boldsymbol{z}_\mathcal{V}, \boldsymbol{z}_\mathcal{S}|\mathcal{V}, \mathcal{S})$ and the true one $p(\boldsymbol{z}_\mathcal{V}, \boldsymbol{z}_\mathcal{S}|\mathcal{V}, \mathcal{S})$. This KL divergence is non-negative, so the second term is the *evidence lower bound (ELBO)* on the log-likelihood $\log p_\theta(\boldsymbol{v}, \boldsymbol{s})$.

Following the derivation in VAE (Kingma & Welling, 2013), when maximizing the above joint log-likelihood, we can maximize the following *ELBO* as:

$$\max_{\theta,\phi} \mathcal{L}_{\text{ELBO}} = \underbrace{\mathbb{E}_{q_\phi(\boldsymbol{z}_\mathcal{V}, \boldsymbol{z}_\mathcal{S}|\mathcal{V}, \mathcal{S})}[\log p_\theta(\boldsymbol{v}, \boldsymbol{s}|\boldsymbol{z}_\mathcal{V}, \boldsymbol{z}_\mathcal{S})]}_{\text{encoder-decoder}} \underbrace{- D_{KL}[q_\phi(\boldsymbol{z}_\mathcal{V}, \boldsymbol{z}_\mathcal{S}|\mathcal{V}, \mathcal{S})||p(\boldsymbol{z}_\mathcal{V}, \boldsymbol{z}_\mathcal{S})]}_{\text{joint prior regularization}}, \tag{11}$$

where $p_\theta(\boldsymbol{v}, \boldsymbol{s}|\boldsymbol{z}_\mathcal{V}, \boldsymbol{z}_\mathcal{S})$ is the conditional distribution parameterized by $\theta$. The first term actually shows an encoder-decoder architecture, while the second term indicates a joint prior regularization on $q_\phi(\boldsymbol{z}_\mathcal{V}, \boldsymbol{z}_\mathcal{S}|\mathcal{V}, \mathcal{S})$.

**The Encoder-Decoder**: Given $\mathcal{V}$ and $\mathcal{S}$, the encoded latent representations $\boldsymbol{z}_\mathcal{V}$ and $\boldsymbol{z}_\mathcal{S}$ are conditional independent, and the posterior can be written as:

$$\begin{aligned} q_\phi(\boldsymbol{z}_\mathcal{V}, \boldsymbol{z}_\mathcal{S}|\mathcal{V}, \mathcal{S}) &= q_{\phi_1}(\boldsymbol{z}_\mathcal{V}|\boldsymbol{z}_\mathcal{S}, \mathcal{V}, \mathcal{S})q_{\phi_2}(\boldsymbol{z}_\mathcal{S}|\mathcal{V}, \mathcal{S}) \\ &= q_{\phi_1}(\boldsymbol{z}_\mathcal{V}|\mathcal{V}, \mathcal{S})q_{\phi_2}(\boldsymbol{z}_\mathcal{S}|\mathcal{V}, \mathcal{S}), \end{aligned} \tag{12}$$

which indicates both the representations of historical items and scenes are not solely dependent from their own sequences. Instead, these representations are derived from $\mathcal{V}$ and $\mathcal{S}$, indicating that the representation of historical items $\boldsymbol{z}_\mathcal{V}$ is influenced by the contextual scene sequence, similarly influencing $\boldsymbol{z}_\mathcal{S}$.

Similarly, given $\boldsymbol{z}_\mathcal{V}$ and $\boldsymbol{z}_\mathcal{S}$, $\boldsymbol{v}$ and $\boldsymbol{s}$ are conditional independent, the conditional distribution is written as:

$$\begin{aligned} p_\theta(\boldsymbol{v}, \boldsymbol{s}|\boldsymbol{z}_\mathcal{V}, \boldsymbol{z}_\mathcal{S}) &= p_{\theta_1}(\boldsymbol{v}|\boldsymbol{s}, \boldsymbol{z}_\mathcal{V}, \boldsymbol{z}_\mathcal{S})p_{\theta_2}(\boldsymbol{s}|\boldsymbol{z}_\mathcal{V}, \boldsymbol{z}_\mathcal{S}) \\ &= p_{\theta_1}(\boldsymbol{v}|\boldsymbol{z}_\mathcal{V}, \boldsymbol{z}_\mathcal{S})p_{\theta_2}(\boldsymbol{s}|\boldsymbol{z}_\mathcal{V}, \boldsymbol{z}_\mathcal{S}), \end{aligned} \tag{13}$$

which indicates we employ both the information from $\boldsymbol{z}_\mathcal{V}$ and $\boldsymbol{z}_\mathcal{S}$ to make the individual prediction of $\boldsymbol{v}$ and $\boldsymbol{s}$.

**The Joint Prior Regularization**: The second term in Eq. 11 represents a joint prior on the posterior $q_\phi(\boldsymbol{z}_\mathcal{V}, \boldsymbol{z}_\mathcal{S}|\mathcal{V}, \mathcal{S})$ for $\boldsymbol{z}_\mathcal{V}$ and $\boldsymbol{z}_\mathcal{S}$. Given the complexity of the joint prior $p(\boldsymbol{z}_\mathcal{V}, \boldsymbol{z}_\mathcal{S})$, we simplify its implementation by assuming $p(\boldsymbol{z}_\mathcal{V}, \boldsymbol{z}_\mathcal{S}) = p(\boldsymbol{z}_\mathcal{V})p(\boldsymbol{z}_\mathcal{S})$. This choice aligns with recent works (Chen et al., 2023b; Tomczak & Welling, 2017), allowing for a more straightforward and efficient implementation. By integrating this with Eq. 12, the joint prior regularization (Eq. 11) can be formulated as:

$$\begin{aligned} D_{KL}[q_\phi(\boldsymbol{z}_\mathcal{V}, \boldsymbol{z}_\mathcal{S}|\mathcal{V}, \mathcal{S})||p(\boldsymbol{z}_\mathcal{V}, \boldsymbol{z}_\mathcal{S})] &= D_{KL}[q_\phi(\boldsymbol{z}_\mathcal{V}|\mathcal{V}, \mathcal{S})q_\phi(\boldsymbol{z}_\mathcal{S}|\mathcal{V}, \mathcal{S})||p(\boldsymbol{z}_\mathcal{V})p(\boldsymbol{z}_\mathcal{S})] \\ &= D_{KL}[q_\phi(\boldsymbol{z}_\mathcal{V}|\mathcal{V}, \mathcal{S})||p(\boldsymbol{z}_\mathcal{V})] + D_{KL}[q_\phi(\boldsymbol{z}_\mathcal{S}|\mathcal{V}, \mathcal{S})||p(\boldsymbol{z}_\mathcal{S})]. \end{aligned} \tag{14}$$

It is worthwhile to point out that the joint prior assumption $p(\boldsymbol{z}_\mathcal{V}, \boldsymbol{z}_\mathcal{S}) = p(\boldsymbol{z}_\mathcal{V})p(\boldsymbol{z}_\mathcal{S})$ is not a perfect choice. In future work, we may explore more intricate joint priors, leveraging the insights from (Tomczak & Welling, 2017; Rezende & Mohamed, 2015; Yin & Zhou, 2018).

**Rewrite the $\mathcal{L}_{\text{ELBO}}$**: By integrating Eq. 12, Eq. 13 and Eq. 14 together, we can rewrite the *ELBO* in Eq. 11 as follows:

$$\begin{aligned} \max_{\theta_1, \theta_2, \phi_1, \phi_2} \mathcal{L}_{\text{ELBO}} = &\mathbb{E}_{q_{\phi_1}(\boldsymbol{z}_\mathcal{V}|\mathcal{V}, \mathcal{S})q_{\phi_2}(\boldsymbol{z}_\mathcal{S}|\mathcal{V}, \mathcal{S})}[\log p_{\theta_1}(\boldsymbol{v}|\boldsymbol{z}_\mathcal{V}, \boldsymbol{z}_\mathcal{S})p_{\theta_2}(\boldsymbol{s}|\boldsymbol{z}_\mathcal{V}, \boldsymbol{z}_\mathcal{S})] \\ &- D_{KL}[q_{\phi_1}(\boldsymbol{z}_\mathcal{V}|\mathcal{V}, \mathcal{S})||p(\boldsymbol{z}_\mathcal{V})] - D_{KL}[q_{\phi_2}(\boldsymbol{z}_\mathcal{S}|\mathcal{V}, \mathcal{S})||p(\boldsymbol{z}_\mathcal{S})]. \end{aligned} \tag{15}$$

Table 3: The statistics of datasets.

| Dataset | #sequences | #items | #scenes | #avg. length | #density |
|---|---|---|---|---|---|
| Outbrain | 46,676 | 238,653 | 3,508 | 2.36 | 9.89e-4% |
| AllScenePay-1m | 1,000,000 | 7,871,700 | 330 | 25.35 | 3.22e-4% |
| AllScenePay-10m | 10,000,000 | 32,766,762 | 801 | 25.33 | 7.70e-5% |

**The Objective Function**: Maximizing the above ELBO is equivalent to minimizing its negative version. We summarize the optimization objective function as:

$$\min_{\theta_1,\theta_2,\phi_1,\phi_2} \mathcal{L} = - \mathbb{E}_{q_{\phi_1}(z_\mathcal{V}|\mathcal{V},\mathcal{S})}[\log p_{\theta_1}(\boldsymbol{v}|\boldsymbol{z}_\mathcal{V},\boldsymbol{z}_\mathcal{S})] - \mathbb{E}_{q_{\phi_2}(z_\mathcal{S}|\mathcal{V},\mathcal{S})}[\log p_{\theta_2}(\boldsymbol{s}|\boldsymbol{z}_\mathcal{V},\boldsymbol{z}_\mathcal{S})]$$
$$+ D_{KL}[q_{\phi_1}(\boldsymbol{z}_\mathcal{V}|\mathcal{V},\mathcal{S})||p(\boldsymbol{z}_\mathcal{V})] + D_{KL}[q_{\phi_2}(\boldsymbol{z}_\mathcal{S}|\mathcal{V},\mathcal{S})||p(\boldsymbol{z}_\mathcal{S})]. \quad (16)$$

where the first and second term indicate we obtain the latent representations from historical sequences $\mathcal{V},\mathcal{S}$, and then we use them to predict the future item $v$ and scene $s$. The third and fourth term show prior regularization on the latent representations, which can be implemented by adversarial learning shown in (Makhzani et al., 2016b). *Therefore, we can see that the objective function above actually is the same as our dual sequence framework without specifying the detailed encoder-decoder networks and prediction loss.*

## B  DETAILS ABOUT ALLSCENEPAY-1M/10M DATASETS

Incorporating scene information for modeling sequential user behavior is a compelling and important area of research in real-world applications. However, this topic has not received extensive attention due to a lack of publicly available datasets for academic purposes. To address this research data gap, we collected 37-day sequential user purchase behaviors from our e-shopping app, covering all scenes (*e.g.* recommendation, text2product search, image2product search, VIPs), and constructed two real-world datasets for academic research.

In particular, the user purchase behaviors range from 2024-07-01 to 2024-08-07, containing over hundreds of millions of users and items. To study the sequential user behavior prediction issue, we take user behaviors between 2024-07-01 and 2024-07-31 as historical behaviors, while behaviors between 2024-08-01 and 2024-08-07 as prediction behaviors. Since the original datasets contain a large number of users having very few purchases, we preprocess the data to enhance its usability while preserving the original real-world behavior distribution. Specifically, we filter out users who have fewer than three historical purchases or fewer than three prediction purchases. After this preprocessing, we still have nearly 50 million user sequences and 100 million items, presenting great challenges in GPU training for academic usage. Thereby, we randomly sample 1 million and 10 million user sequences as the AllScenePay-1m and AllScenePay-10m dataset.

The two datasets encompass users' purchase behaviors across all scenes in our app from July 1, 2024, to August 7, 2024. Each user's purchase activities form a sequence data. Each sequence data includes seven feature fields "user_id", "history_item_ids", "history_scene_ids","history_timestamps", "future_item_ids", "future_scene_ids", "future_timestamps"[4]. All feature fields except the timestamp are hashed for anonymization. One example is given as follows:

```
user_id: 0
history_item_ids: 12,32,3,90,7
history_scene_ids: 293,43,53,23,11
history_timestamps: 20240701,20240701,20240708,20240721,20240721
future_item_ids: 9,101,35
future_scene_ids: 73,91,137
future_timestamps: 20240802,20240807,20240807
```

The general statistics of these two datasets are shown in Table 3. From this table, we can see that both two datasets are extremely sparse. The number of items and scenes are quite large, bringing

---

[4]In our work, we just use the user_id to identify different sequences in data processing and do not use it as input feature. This helps the model to handle the user-cold-start issue, and we can make predictions of arbitrary users as long as the historical item purchases are given.

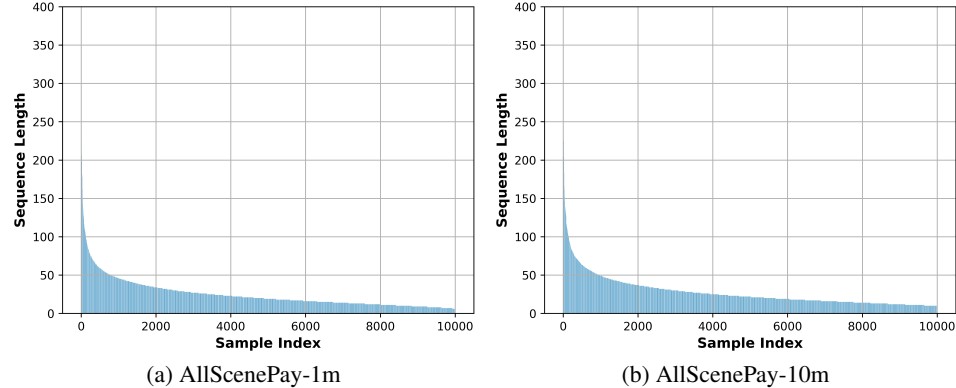

(a) AllScenePay-1m  (b) AllScenePay-10m

Figure 5: The long-tailed distribution of sequence length of two industrial datasets. This figure illustrates that the majority of users engage in only a limited number of interactions, presenting modeling challenges of user behaviors.

Table 4: Performance comparison of different methods on period item prediction task. R@$k$ and N@$k$ represent Recall@$k$ and NDCG@$k$, respectively. We use "w/o" to denote DSPnet without a particular part. The best results are bolded and the most competitive results are underlined.

| Dataset | Outbrain | | | | AllScenePay-1m | | | | AllScenePay-10m | | | |
|---|---|---|---|---|---|---|---|---|---|---|---|---|
| Method | R@5 | N@5 | R@10 | N@10 | R@5 | N@5 | R@10 | N@10 | R@5 | N@5 | R@10 | N@10 |
| BERT4Rec | 0.0946 | 0.0694 | 0.1388 | 0.084 | OOM | OOM | OOM | OOM | OOM | OOM | OOM | OOM |
| MSDP | 0.2671 | 0.2191 | 0.2967 | 0.229 | 0.0007 | 0.0006 | 0.0011 | 0.0008 | 0.0008 | 0.0007 | 0.0012 | 0.0009 |
| ContraRec | 0.3573 | 0.2481 | 0.4654 | 0.2844 | 0.0759 | 0.0790 | 0.0888 | 0.0839 | 0.1534 | 0.1610 | 0.1796 | 0.1707 |
| SceneCTC | 0.4754 | 0.4080 | 0.5184 | 0.4225 | 0.0758 | 0.0785 | 0.0913 | 0.0848 | 0.1538 | 0.1585 | 0.1813 | 0.1694 |
| SceneContraRec | 0.4902 | 0.4031 | 0.5388 | 0.4197 | 0.0790 | 0.0821 | 0.0933 | 0.0876 | 0.1532 | 0.1587 | 0.1820 | 0.1702 |
| CARCA | 0.5092 | 0.4393 | 0.5402 | 0.4499 | OOM | OOM | OOM | OOM | OOM | OOM | OOM | OOM |
| DSPnet– | 0.5281 | 0.4634 | 0.5570 | 0.4372 | 0.0867 | 0.0905 | 0.1029 | 0.0968 | 0.1636 | 0.1703 | 0.1922 | 0.1812 |
| DSPnet(w/o $\mathcal{L}_{\text{APR}}$, $\mathcal{L}_{\text{CCR}}$) | 0.6062 | 0.5307 | 0.6583 | 0.5482 | 0.0997 | 0.1053 | 0.1121 | 0.1090 | 0.1895 | 0.1998 | 0.2149 | 0.2083 |
| DSPnet(w/o $\mathcal{L}_{\text{CCR}}$) | 0.6069 | 0.5347 | 0.6635 | 0.5537 | 0.0995 | 0.1049 | 0.1120 | 0.1088 | 0.1930 | 0.2033 | 0.2186 | **0.2118** |
| DSPnet(w/o $\mathcal{L}_{\text{APR}}$) | 0.6151 | **0.5396** | 0.6651 | **0.5562** | 0.1007 | 0.1060 | 0.1148 | 0.1107 | **0.1930** | **0.2037** | 0.2178 | 0.2117 |
| DSPnet | **0.6198** | 0.5388 | **0.6682** | 0.5549 | **0.1015** | **0.1071** | **0.1149** | **0.1114** | 0.1926 | 0.2028 | **0.2187** | 0.2115 |
| | (+11.06%) | (+9.95%) | (+12.80%) | (+10.50%) | (+2.25%) | (+2.50%) | (+2.16%) | (+2.38%) | (+3.88%) | (+4.18%) | (+3.67%) | (+4.08%) |

challenges in modeling user behaviors. Further, we randomly sample 10,000 sequences and make an analysis about the sequence length distribution. The results are given in Figure 5.

## C   MODEL COMPLEXITY COMPARISON

We analyze the complexity of different models via two components: feature encoding and behavior prediction, both of which are commonly present in sequential behavior prediction models. The compared methods all use the powerful and popular transformer encoder. Let $B$ be the batch size, $L$ be the number of transformer layers, $|\mathcal{T}|$ be the sequence length of samples, $H$ be the head number and $d$ be the dimension of each head. The time complexity of transformer encoder can be represented as $O(B * L * H * |T|^2 * d)$, which is nearly the same for all compared methods. The main difference of complexity lies in behavior prediction. Before analyzing the complexity of behavior prediction part, we denote $K^v$ as the number of candidate items (including positive and negative ones) for prediction, the complexity comparison is listed in Table 5.

In this table, $K^v$ generally reaches the magnitude of millions in industrial settings. The value of $|\mathcal{T}|$ varies depending on the dataset, and for our business applications, we typically set it to 100. Consequently, the complexity of DSPnet is considerably lower than that of CARCA and Bert4Rec and does not significantly increase over SOTA methods such as SceneContraRec. This makes DSP-

Table 5: Time complexity of different models on behavior prediction part.

| Method | Behavior Prediction | Remark |
|---|---|---|
| Bert4Rec | $O(B * \rho * |\mathcal{T}| * K^v)$ | $\rho$ is the ratio of sequence tokens for Cloze task |
| CARCA | $O(B * K^v + B * H' * (K^v * N') * d)$ | It involves cross attention between user-side features and candidate item. $H'$ is number of heads in cross attention, and $N'$ is the number of user-side features |
| SceneCTC | $O(B * K^v)$ | It has no contrastive loss |
| MSDP | $O(B * K^v + B^2 * d)$ | It involves contrastive loss of input sequence |
| ContraRec | $O(B * K^v + B^2 * d)$ | It involves contrastive loss of input sequence |
| SceneContraRec | $O(B * K^v + B^2 * d)$ | It involves contrastive loss of input sequence |
| DSPnet | $O(B * K^v + 2 * B^2 * d)$ | It involves CCR loss of two input sequences |

(a)Outbrain  (b)Outbrain  (c)AllScenePay-1m  (d)AllScenePay-1m

Figure 6: The performance of different hyper-parameters.

Table 6: Study of the prior distribution.

| Dataset | Ourbrain | | | | AllScenePay-1m | | | |
|---|---|---|---|---|---|---|---|---|
| Model | R@5 | N@5 | R@10 | N@10 | R@5 | N@5 | R@10 | N@10 |
| Uniform | 0.6025 | 0.5190 | 0.6537 | 0.5355 | 0.0851 | 0.0621 | 0.1129 | 0.0711 |
| Laplace | 0.5867 | 0.5031 | 0.64 | 0.5206 | 0.0839 | 0.0610 | 0.1127 | 0.0703 |
| Multi-Gaussian | 0.5982 | 0.5181 | 0.6485 | 0.5343 | 0.0842 | 0.0618 | 0.1114 | 0.0705 |
| Lognormal | 0.6177 | 0.5373 | 0.6648 | 0.5525 | 0.0843 | 0.0615 | 0.1131 | 0.0708 |
| Standard Guassian | **0.6248** | **0.5368** | **0.6717** | **0.5520** | **0.0870** | **0.0632** | **0.1155** | **0.0725** |

net well-suited for usage in large-scale industrial datasets. We have successfully deployed it in our system using 16 A100 GPUs.

# D    MORE EXPERIMENTS

## D.1    HYPER-PARAMETER SENSITIVITY

In our DSPnet, $\alpha$ and $\beta$ control the weight on adversarial prior regularization and conditional contrastive regularization, respectively. We here investigate the effects on performance of these two hyper-parameters. The results are collected in Figure 6.

From this figure, when comparing different columns, we see that the model performance varies a lot, highlighting the substantial impact of the loss weight on CCR. Additionally, the optimal hyper-parameter settings differ, primarily due to the distinct data distributions of Outbrain and AllScenePay-1m. It is practical to set these hyper-parameters according to the used data.

## D.2    EFFECTS OF DIFFERENT PRIORS

DSPnet incorporates adversarial prior regularization loss to impose prior knowledge on the learned representations. In this part, we explore the influence of different prior distributions on model performance. In particular, we use the uniform distribution $\mathcal{U}(0,1)$, the Laplace distribution $\text{Laplace}(0,1)$, the standard normal distribution $\mathcal{N}(0,1)$, and the lognormal distribution $\text{Lognormal}(0,1)$. Additionally, we used a sum of two normal distributions, $\mathcal{N}(0,1) + \mathcal{N}(3,1)$, to construct Multi-Gaussian distribution. The results are provided in Table 6.

By analyzing the results from this table, we see that the standard Gaussian distribution consistently shows the best performance on three datasets. This observation matches a widely accepted principle in recommendation and search systems, where user preferences tend to exhibit Gaussian distribution(Liang et al., 2018; Cui et al., 2018; Xie et al., 2021). While a Multi-Gaussian approach has the potential to capture user preferences more accurately, given that individuals often have multiple areas of interest, determining the parameters for a Multi-Gaussian model can be rather complex. Consequently, employing the standard Gaussian serves as a straightforward and effective choice of leveraging prior knowledge in practical applications.

## D.3    ON THE PREDICTION PERFORMANCE OF SCENE

We conducted additional experiments to evaluate the scene prediction capabilities of various models, aiming to demonstrate that our DSPnet more effectively captures the "scene" aspect alongside item sequences. The results, presented in Table 7, indicate that DSPnet achieves better scene prediction ability, validating the effectiveness of our idea.

Table 7: Performance comparison of different methods on next scene prediction task. R@$k$ and N@$k$ represent Recall@$k$ and NDCG@$k$, respectively.

| Dataset | Outbrain | | | | AllScenePay-1m | | | |
|---|---|---|---|---|---|---|---|---|
| Method | R@5 | N@5 | R@10 | N@10 | R@5 | N@5 | R@10 | N@10 |
| BERT4REC | 0.3043 | 0.2763 | 0.3345 | 0.2861 | 0.8253 | 0.6365 | 0.9205 | 0.6483 |
| MSDP | 0.2638 | 0.1798 | 0.3789 | 0.2167 | 0.8464 | 0.6489 | 0.9371 | 0.6788 |
| ContraRec | 0.4071 | 0.3468 | 0.4662 | 0.3661 | 0.8710 | 0.6635 | 0.9508 | 0.6903 |
| SceneCTC | 0.6246 | 0.5492 | 0.6937 | 0.5715 | 0.8872 | 0.6692 | 0.9547 | 0.6918 |
| SceneContraRec | 0.6175 | 0.5386 | 0.6858 | 0.5606 | 0.8692 | 0.6629 | 0.9528 | 0.6911 |
| DSPnet | **0.6567** | **0.5770** | **0.7200** | **0.5975** | **0.8944** | **0.6816** | **0.9629** | **0.7045** |

