# OpenReview forum: "Predicting User Behaviors with Scene via Dual Sequence Networks"
_ICLR.cc/2025/Conference — Submitted to ICLR 2025_

### Official Review · Reviewer_od31 · 2024-10-31

**Soundness:** 2
**Presentation:** 3
**Contribution:** 2
**Rating:** 3
**Confidence:** 4

**Summary:**

The paper focuses on using scene/behavior features to enhance sequential recommendation. The authors design a Dual Sequence Prediction network (DSPNet), constructing parallel item sequence encoders along with corresponding scene sequence encoders to form a dual-branch structure. They employ an improved contrastive learning approach to enhance the robustness of both sequence models, utilize adversarial loss for cross-branch alignment, and train the model by combining "next item prediction" with "next corresponding behavior prediction." The effectiveness of the proposed DSPNet is validated on one public dataset and two internal datasets.

**Strengths:**

1. Using behavior type features as contextual information indeed enhances the effectiveness of sequential recommendation.
2. The paper is well-written and very easy to understand.
3. It includes experiments on industrial-scale datasets.

**Weaknesses:**

1. **Limited novelty**. The effectiveness of using behavior features as contextual features in personalized recommendations is somewhat a consensus, so emphasizing this appears somewhat trivial. The proposed method is essentially a simple combination of existing practices, including contrastive learning, adversarial training, next-item, and next-scene/behavior prediction, which are lukewarm in this domain. Although a marginal technical contribution is made with conditional weights in contrastive learning, its superiority over standard contrastive loss does not seem supported by the experiments. While some theoretical results are provided, such as Lemma 1 proving that simultaneously predicting the next item and its corresponding scene/behavior minimizes ELBO, this does not appear significant enough to suffice as a novel technical contribution.

2. **Inconsistency between the dual-branch approach and motivation**. Modeling item sequences and scene/behavior sequences with dual branches seems disconnected or even in conflict with the stated motivation. Lines 93-94 mention that item and scene *simultaneously* occur in Figure 1(c), yet the method here appears to overlook this point. This presents a contradiction—how can independent modeling of item and scene sequences ensure their one-to-one correspondence? It is unclear whether the proposed **coarse-grained** sequence-level alignment/fusion is superior to a **fine-grained** alignment/fusion between item-scene/behavior pairs.

3. **Insufficient experimental comparisons**. The paper lacks crucial baselines that would support the superiority of the proposed method, such as: (1) self-supervised sequential recommendation models, like SASRec, S3-Rec [1], ICLRec [2], and DCRec [3]; (2) baselines using scene/behavior features as attributes [4,5]. Additionally, a simple baseline could involve adding scene/behavior embeddings to item embeddings and using straightforward models like SASRec or BERTRec, often considered strong baselines in practice; (3) multi-behavior sequential recommendation models [6-11] and multi-behavior recommendation models [12-15].

4. **Lack of context in related work**. This study's application is highly relevant to multi-behavior recommendations, particularly multi-behavior sequential recommendations [16], yet these works are overlooked without any discussion, reflecting a gap in the paper's background and depth.


--------
***References***

[1] Zhou K, Wang H, Zhao W X, et al. S3-rec: Self-supervised learning for sequential recommendation with mutual information maximization[C]//Proceedings of the 29th ACM international conference on information & knowledge management. 2020: 1893-1902.

[2] Chen Y, Liu Z, Li J, et al. Intent contrastive learning for sequential recommendation[C]//Proceedings of the ACM Web Conference 2022. 2022: 2172-2182.

[3] Yang Y, Huang C, Xia L, et al. Debiased contrastive learning for sequential recommendation[C]//Proceedings of the ACM web conference 2023. 2023: 1063-1073.

[4] Wu L, Li S, Hsieh C J, et al. SSE-PT: Sequential recommendation via personalized transformer[C]//Proceedings of the 14th ACM conference on recommender systems. 2020: 328-337.

[5] Rashed A, Elsayed S, Schmidt-Thieme L. Context and attribute-aware sequential recommendation via cross-attention[C]//Proceedings of the 16th ACM Conference on Recommender Systems. 2022: 71-80.

[6] Elsayed S, Rashed A, Schmidt-Thieme L. Multi-Behavioral Sequential Recommendation[C]//Proceedings of the 18th ACM Conference on Recommender Systems. 2024: 902-906.

[7] Su J, Chen C, Lin Z, et al. Personalized behavior-aware transformer for multi-behavior sequential recommendation[C]//Proceedings of the 31st ACM International Conference on Multimedia. 2023: 6321-6331.

[8] Xia L, Huang C, Xu Y, et al. Multi-behavior sequential recommendation with temporal graph transformer[J]. IEEE Transactions on Knowledge and Data Engineering, 2022, 35(6): 6099-6112.

[9] Yang Y, Huang C, Xia L, et al. Multi-behavior hypergraph-enhanced transformer for sequential recommendation[C]//Proceedings of the 28th ACM SIGKDD conference on knowledge discovery and data mining. 2022: 2263-2274.

[10] Yuan E, Guo W, He Z, et al. Multi-behavior sequential transformer recommender[C]//Proceedings of the 45th international ACM SIGIR conference on research and development in information retrieval. 2022: 1642-1652.

[11] Liu Z, Hou Y, McAuley J. Multi-Behavior Generative Recommendation[C]//Proceedings of the 33rd ACM International Conference on Information and Knowledge Management. 2024: 1575-1585.

[12] Jin B, Gao C, He X, et al. Multi-behavior recommendation with graph convolutional networks[C]//Proceedings of the 43rd international ACM SIGIR conference on research and development in information retrieval. 2020: 659-668.

[13] Xuan H, Liu Y, Li B, et al. Knowledge enhancement for contrastive multi-behavior recommendation[C]//Proceedings of the sixteenth ACM international conference on web search and data mining. 2023: 195-203.

[14] Xia L, Xu Y, Huang C, et al. Graph meta network for multi-behavior recommendation[C]//Proceedings of the 44th international ACM SIGIR conference on research and development in information retrieval. 2021: 757-766.

[15] Wu Y, Xie R, Zhu Y, et al. Multi-view multi-behavior contrastive learning in recommendation[C]//International conference on database systems for advanced applications. Cham: Springer International Publishing, 2022: 166-182.

[16] Chen X, Li Z, Pan W, et al. A Survey on Multi-Behavior Sequential Recommendation[J]. arXiv preprint arXiv:2308.15701, 2023.

**Questions:**

1. In Table 1, why does BERT4Rec encounter 'OOM' while the other models do not?

2. During inference, how are the metrics calculated? Besides considering the accuracy of item prediction, was the accuracy of behavior prediction also considered?

---

> ### Author Response · Authors · 2024-11-24
> **Response to Reviewer od31**
>
> `W1 and W4:`
> **There is a misunderstanding between our scene feature and multi-behavior feature, so we first show their differences and then introduce the novelty of our work.**
>
> ***The Difference:***  **Scene features are defined as sub-interfaces within an app or website, created by designers to achieve specific functionalities.** In Figure 1(a) of the paper, an e-shopping app has many sub-interfaces with each has different functionalities such as text search and image search. Text search enables users to use text query for buying products, while image search enables users to use image query for buying products. **In contrast, behavior-type features (e.g., view, click, add to cart, pay) are generated based on user interactions.** Modeling the scene and multiple behaviors are two different problems.** In our model and experiments, we only consider one-type behavior**, *i.e.*, ''view'' in Outbrain and ''pay'' in our industrial datasets. As a result, it is not appropriate to compare DSPnet with multiple behavior sequential RS works.
>
> ***The Novelty:*** **Integrating the defined scene features into sequential behavior modeling constitutes an important and new problem setting derived from our industrial business.** Currently, there are limited studies addressing this area because of inaccessible data. Existing scene-based sequential RS [16-19] employ different scene definitions from ours. In Figure 1(b), we have shown that these scene features significantly impact user engagements. So, we propose a new framework specifically designed to incorporate the scene feature in sequential behavior modeling. **DSPnet consists of two main and original components: dual sequence learning with sequence feature enhancement and CCR loss, with both are designed to tackle important challenges.** The first one is to effectively encode sequential dynamics and deliver these dynamics to scene and item side for predicting behaviors. The second one is to learn representation invariance and improve the model's robustness against skewness. **Recognizing the research data gap in this practical area, we intend to release our collected datasets to the community.**
>
> `W2:`
> **We do not agree that our design presents a contradiction with our motivation. Our design is to better capture the scene-item correspondence in the context of random and noisy user behaviors.** Specifically, user behaviors often contain noise and randomness at single behavior level, whereas behaviors at sequence level provide more stable user dynamics. **In order to better capture user dynamics and mitigate temporal errors, we preserve the correspondence at sequence level in our dual sequence encoder.** Additionally, **DSPnet establishes the one-to-one correspondence by the joint prediction objective**, where each scene-item pair is predicted simultaneously. By adjusting the time step $T$ of sequences, multiple (scene, item) pairs can be consistently predicted in the sequence. **Our results and theoretical analysis show the superiority of our design.
>
> `W3:`
> (1) We did experiments of SASRec and S3-Rec in the beginning, whose performance are not good. So we only compared to models with more advanced encoders and supervision signals. Also, we have added the results of CARCA in the following table. **CARCA contains a cross attention module that easily leads to OOM on large-scale datasets.**
>
> | Method | Outbrain R@5 | Outbrain N@5 | Outbrain R@10 | Outbrain N@10 | AllScenePay-1m R@5 | AllScenePay-1m N@5 | AllScenePay-1m R@10 | AllScenePay-1m N@10 |
> |---|---|---|---|---|---|---|---|---|
> | **CARCA**  | 0.5126       | 0.4373       | 0.5430        | 0.4472        | OOM                | OOM                | OOM                 | OOM                 |
> | **DSPnet** | 0.6248       | 0.5368       | 0.6717        | 0.5520        | 0.0870             | 0.0632             | 0.1155              | 0.0725              |
>
> (2) We originally considered using scene feature as attributes to perform as strong baselines, like the results of SceneCTC and SceneContraRec in Table 1 of our original paper. We further added the suggested baseline CARCA.
>
> (3) and (4) The scene feature defined in this paper is fundamentally different from multi-behavior feature, representing two distinct problems. Our method and experiments use only one type behavior, so it is not appropriate to compare DSPnet with multi-behavior sequential RS.
>
> `Q1:`
> BERT4Rec adopts the Cloze task, which produces an output shaped `[batch_size, sequence_length, num_candidate_items]`. While others typically output `[batch_size, num_candidate_items]`.  In our industrial datasets, there are over 7 million candidate items and sequences can be up to 100 items long. This makes BERT4Rec have OOM errors, even using A100 GPUs.
>
> `Q2:`
> In inference, our single-type behavior method predicts the next item a user may interact with from a candidate set and evaluates performance using Recall@k and NDCG@k metrics.

---

> > ### Comment · Reviewer_od31 · 2024-11-24
> > **Post-rebuttal**
> >
> > Thank you for your response. Unfortunately, my primary concerns remain unaddressed. Therefore, I decide to maintain my score.
> >
> > 1. I appreciate the authors’ efforts to clarify the difference between scene features and behavior features, i.e., they come from different sources.
> >    However, regarding their usage, both features can be categorized as attributes and considered side information of the item. From a business perspective, there might be subtle differences, but technically, they are equally compatible with existing multi-behavior recommendation methods.
> >    As Reviewer 8nu5 also mentioned, I believe that a more technical discussion of related work is essential. Simply ignoring this conceptual overlap is not advisable, as it could severely limit the scope and significance of the paper.
> >    Similarly, to highlight the technical contributions and effectiveness of DSPNet, the aforementioned experimental comparisons are indispensable.
> >
> > 2. Essentially, the design of DSPNet neglects the one-to-one correspondence within the sequence side (i.e., the simultaneous occurrence of items and scenes), which might introduce unfavorable biases. Furthermore, whether the proposed coarse-grained sequence-level alignment/fusion is superior to a fine-grained alignment/fusion between item-scene pairs is unclear. The authors may need to supplement additional comparative experiments to demonstrate the superiority of the former. Intuitively, the latter seems more reasonable.
> >
> > The authors are encouraged to carefully reconsider the reviewers’ comments, as this could further improve the quality of the paper.

---

> ### Author Response · Authors · 2024-11-25
> **Response to Reviewer od31**
>
> `Q1`:
> Thanks for the response. **DSPnet has both relations and differences with the listed Multi-Behavior Sequential RS (MBSRS)  works.**
>
> `The Relation:` MBSRS is a significant challenge because it involves modeling both **multi-behaviors** and **behavioral sequences**, to consider the existing problems of Multi-Behavior RS and Single-Behavior Sequential RS [9]. **In this work, we focus on the sub-problem, incorporating scene features into Single-Behavior Sequential RS.**
>
> `Technical Difference:` **MBSRS generally use a one-to-one correspondence encoding between behavior types and items [3-8].** However, our research emphasizes that randomness and noise in user behaviors can introduce misalignment issue between scenes and items (an example is given in following response). To address this issue, **we developed a dual sequence encoding approach for sequence-level correspondence in encoding.** This approach simultaneously learns reliable user dynamics against misaligned scene-item pairs and delivers the dynamics to both scene and item sides for behavior prediction.
>
> `Result Difference:` **Our sequence-level correspondence encoding approach outperforms the existing one-to-one correspondence encoding style.** Table 1 in our paper compares our method with popular Single-Behavior Sequential RS methods (*e.g.*, CARCA, SceneCTC, SceneContraRec) that use the one-to-one correspondence encoding. **Additionally, in the following table, we tested replacing our dual sequence encoding with the one-to-one encoding in existing works [3-8].**
>
> **Table: Performance comparison of DSPnet-- and our DSPnet. DSPnet-- means we replace our dual sequence encoder with one-to-one correspondence encoding in recent multi-behavior sequential RS works.**
> | Dataset           | Method  | R@5    | N@5    | R@10   | N@10   |
> |---|---|---|---|---|---|
> | **Outbrain**      | DSPnet-- | 0.5324 | 0.4612 | 0.5604 | 0.4703 |
> |                   | **DSPnet** | 0.6248 | 0.5368 | 0.6717 | 0.5520 |
> | **AllScenePay-1m**| DSPnet-- | 0.0742 | 0.0527 | 0.1047 | 0.0625 |
> |                   | **DSPnet** | 0.0870 | 0.0632 | 0.1155 | 0.0725 |
>
> *We will add this discussion, results and the related MBSRS works in the revised version.*
>
> `Q2`:
> **The one-to-one correspondence of scenes and items within the sequence may face the misalignment problem.** For example, in an e-shopping app, a user might see certain products in the "recommendation" interface with the intention to purchase them. However, instead of buying immediately, the user adds these products to the "cart" interface and buys from the "cart" interface later, because of upcoming sales promotions from sellers. In this case, the user's interest and intent are initially reflected in the "recommendation" interface but are incorrectly collected in the ''cart'' interface.
>
> **To this end, we establish correspondence at the sequence level, enabling the model to effectively address fine-grained misalignment issues.** We also conducted an experiment in which we replaced our dual sequence encoder with a one-to-one correspondence encoder [3-8] that processes concatenated scene-item embeddings as a single input. The results of this experiment are presented in the table above. **These findings demonstrate that our dual sequence-level encoder provides superior representation learning capabilities compared to the one-to-one correspondence encoder.**
>
> **We attribute this superior performance to two factors:**
> 1) The one-to-one correspondence encoder could be sensitive to misalignment errors, while our dual sequence-level encoder learn representations against this misalignment.
> 2) The one-to-one correspondence encoder struggles to effectively capture the mutual interactions between scene and item when predicting behaviors. In contrast, our dual sequence encoder incorporates sequence feature enhancement module to explicitly captures these complex relationships, learning better user dynamics for behavior prediction.
>
> **Reference**
> [1] Robust User Behavioral Sequence Representation via Multi-scale Stochastic Distribution Prediction, CIKM'2023
> [2] Sequential Recommendation with Multiple Contrast Signals, TOIS'2023
> [3] Personalized behavior-aware transformer for multi-behavior sequential recommendation, MM 2023.
> [4] Multi-behavior sequential recommendation with temporal graph transformer, TKDE 2022.
> [5] Multi-behavior hypergraph-enhanced transformer for sequential recommendation, KDD 2022
> [6] Multi-behavior sequential transformer recommender, SIGIR 2022.
> [7] Knowledge enhancement for contrastive multi-behavior recommendation, WSDM 2023
> [8] Multi-view multi-behavior contrastive learning in recommendation, International conference on database systems for advanced applications, 2022.
> [9] A Survey on Multi-Behavior Sequential Recommendation. arXiv 2023

---

> ### Author Response · Authors · 2024-11-27
> **Response to reviewer od31**
>
> Dear Reviewer od31, we sincerely appreciate your valuable suggestions and comments. As the comment phase is drawing to a close, we would like to know if all of your concerns have been adequately addressed. If you have any further questions or require additional clarification on any aspect of our work, we would be glad to provide thorough responses.

---

### Official Review · Reviewer_8nu5 · 2024-11-03

**Soundness:** 2
**Presentation:** 2
**Contribution:** 3
**Rating:** 5
**Confidence:** 3

**Summary:**

This paper introduces the Dual Sequence Prediction network (DSPnet) to improve future user action prediction by modeling interdependencies between "scene" features (like “text2product search”) and item sequences. This paper claimed that traditional models often overlook these dynamics, leading to potential information loss. DSPnet addresses this by using parallel networks to capture scene and item dependencies and incorporates Conditional Contrastive Regularization (CCR) loss to handle sequence noise. The approach shows enhanced robustness and effectiveness on both public and industrial datasets.

**Strengths:**

(1) This paper proposes a unique Dual Sequence Prediction network (DSPnet) that explicitly models both scene and item sequences. By focusing on the interdependencies between scenes and items, DSPnet captures a level of contextual nuance that is often overlooked in traditional sequential behavior models. This dual-focus approach is well-aligned with real-world applications where user behaviors are influenced by both the content itself and the surrounding context.
(2) Moreover, the introduction of CCR loss is a good point, as it addresses the noise and randomness inherent in sequential user behaviors. By focusing on the invariance of similar historical sequences, CCR loss enhances the model robustness and improves its ability to generalize to diverse, noisy data. This addition is valuable in real-world applications, where user behavior data can be unpredictable.
(3) The dataset may be valuable for the future research if it will be released as promised.

**Weaknesses:**

**Unclear Definition and Scope of "Scene"**: The paper does not clearly define what constitutes a "scene" in the context of user behavior modeling. While scenes are described as features crafted by app or website designers (e.g., “text2product search” and “recommendation”), the criteria for selecting or categorizing these scenes remain ambiguous. It is also unclear how many types of scenes the model considers and whether these categories are generalizable to various platforms or domain-specific. A more precise definition would help readers understand the breadth of the model's applicability and the nature of the contextual features being leveraged.

**Limited Comparison with Multi-Behavior Models**: If scenes are understood as representations of multi-behavior patterns, DSPnet could benefit from comparison with established multi-behavior modeling approaches, such as the work by Cho et al. on Dynamic Multi-Behavior Sequence Modeling for Next Item Recommendation (AAAI 2023). Multi-behavior models aim to capture diverse user actions within sequences, which aligns with the paper’s goal to model interdependencies between scenes and items. A direct comparison could illustrate DSPnet’s advantages (or limitations) over these approaches in capturing the nuances of dynamic user behavior.

**Lack of Standard Metrics for Evaluation**: In evaluating DSPnet on the OutBrain dataset, the paper does not include Mean Average Precision @12, a widely recognized metric for this dataset. Including this metric would facilitate a more transparent and standardized assessment of DSPnet’s performance, allowing comparisons with other models that have used this benchmark. This would also provide a clearer view of DSPnet's efficacy across different evaluation metrics relevant to the field.

**Omission of Scene Prediction Accuracy**: Although DSPnet is designed to jointly predict both scenes and items, the accuracy of its scene prediction component is not reported. Providing this accuracy would clarify how effectively the model captures the contextual "scene" aspect in addition to item sequence modeling. Without this metric, it is challenging to evaluate the effectiveness of DSPnet’s dual-stream approach in fully capturing the dependencies between scenes and items.

**Limited Citation of Recent Works**: The paper’s references do not include any work from 2024 or later, which may indicate that it has not incorporated the most recent advancements in the field. This omission is notable for an ICLR 2025 submission, as the field of sequential user behavior modeling is rapidly evolving. Including more recent literature would strengthen the theoretical grounding of DSPnet and ensure that it is evaluated within the current landscape of user behavior modeling techniques.

**Commonality of Dual-Stream Architectures**: The paper presents DSPnet’s dual-stream architecture, which separately models scene and item sequences, as a novel approach. However, dual-stream architectures have become relatively common for tasks with multiple related objectives, such as multi-task learning frameworks. The paper would benefit from a deeper exploration of how DSPnet’s dual-stream setup is unique or offers advantages over other dual-stream or multi-task models. Additional insights on why DSPnet’s architecture is particularly suited to user behavior prediction would highlight its contributions more effectively.

**Questions:**

(1) How many types of the scenes does this paper consider?
(2) Please justify the lack of recent works and the comparison with multi-behavior recommendation.

---

> ### Author Response · Authors · 2024-11-24
> **Response to Reviewer 8nu5**
>
> `W1, W2 and Q2:`
> Thanks for your suggestion, we give the definition of scene as follows:
> **In online service, scene, created by app or web designers, is the sub-interfaces containing specific themes or functionalities within an App or website.** In Figure 1 (a) of our paper, an e-shopping app has many sub-interfaces with each has different functionalities such as ''text search'' and ''image search''. Text search enables users to use text query for buying products, while image search enables users to use image query for buying products. In Figure 1 (b) of our paper, we conclude that items in different scenes have varied themes and largely influence users' engagements. **So, we can see the ''scene'' feature widely exists in real-world applications, and it can be generalized to various platforms.**
>
> **We emphasize that scene feature is totally different from the widely studied multiple behaviors (*e.g.* view, click, cart, pay). ''Scene'' is created by app or web designers, while multiple behaviors are generated by users.** How to model the scene and multiple behaviors are two different problems. In our model and experiments, we only consider one-type behavior, *i.e.*, ''view'' in Outbrain and ''pay'' in our industrial datasets. It is not appropriate to compare DSPnet with multiple behavior sequential RS works.
>
> `W3:`
> The Outbrain dataset was initially developed for click-through rate prediction, using mean average precision as an evaluation metric. In contrast, this paper concentrates on predicting sequential user behavior within scene feature. In this case, evaluation metrics such as Recall@k and NDCG@k are widely recognized and used [1–6].
>
> After considerable efforts of identifying suitable public datasets, we found that only the Outbrain dataset could meet our requirements. Given the scarcity of available data on this problem, we are going to release our collected industrial datasets and put forward this research topic in the community.
>
> `W4:`
> Thank you for your suggestion. In our industrial case, we usually focus on forecasting future items and thus only present these item prediction results in the paper. **Following your advice, we have done scene prediction. Because of limited response space, we show the results in Appendix-Table 7 of the submitted revised version. From the table, we see that DSPnet consistently outperforms recent models in scene prediction performance.**
>
> `W5 and Q2:`
> Based on the data analysis, we identified the scene feature as a critical factor influencing future behavior occurrences. **To the best of our knowledge, studying how to better employ this scene feature is an important and new problem setting in sequential behavior modeling.** This area remains underexplored primarily due to the lack of available research data. Consequently, we do not elaborate more works on this research problem. **It is important to note that existing scene-based sequential recommendation systems [16-19] employ different scene definitions from ours.** [16] explores adaptive sequential RS across different domains (*e.g.* book, music). [17] defines scene as a collection of predefined item categories. [18] investigates the use of large language models (LLMs) for real-time sequential RS. [19] refers to scenes as 200 predefined topics, such as "weekend spring outing".
>
> Meanwhile, we have also added works published in 2024, including LLMs[5, 6, 7, 8], diffusion models[9, 10, 11] and transformer efficiency [12, 13] for sequential RS .
>
> `W6:`
> We agree that modeling scene sequences can serve as an auxiliary task to enhance item prediction, but DSPnet is specifically designed to integrate scene information into sequence behavior modeling. **This specialized design offers advantages over other dual-stream or multi-task learning (MTL) models.**
>
> Popular MTL frameworks [14,15] primarily focus on modeling task relationships and utilize various parameter-sharing strategies to address the gradient conflict issue. However, these approaches have two main shortcomings on our problem. First, they are unable to capture the mutual effects between scenes and items when predicting future behaviors. Second, they do not account for the randomness, noise, and skewness commonly present in user behavior sequences.
>
> In contrast, DSPnet proposes dual sequence learning with sequence feature enhancement module to **effectively encode sequential dynamics and deliver these dynamics to both scene and item side for predicting behaviors.** Additionally, it introduces CCR loss to **achieve representation invariance and improve the model's robustness on skewed user behavior sequences.**
>
> *We will add this discussion in the new version.*
>
> `Q1:`
> The number of used scenes is given in Appendix-Table 3. In Outbrain, we take the domain name of website as the scene feature. In industrial datasets, we take the ''text search'', ''image search'' and many sub-interfaces within the app as scene feature.

---

> ### Author Response · Authors · 2024-11-27
> **Response to reviewer 8nu5**
>
> Dear Reviewer 8nu5, we sincerely appreciate your valuable suggestions and comments. As the comment phase is drawing to a close, we would like to know if all of your concerns have been adequately addressed. If you have any further questions or require additional clarification on any aspect of our work, we would be glad to provide thorough responses.

---

### Official Review · Reviewer_FSwH · 2024-11-03

**Soundness:** 3
**Presentation:** 3
**Contribution:** 3
**Rating:** 5
**Confidence:** 2

**Summary:**

The paper addresses the challenge of predicting future user actions by effectively modeling sequential user behaviors with a focus on integrating contextual information, particularly the scene feature. Scene features, designed by app or website developers, significantly influence user engagement and exhibit distinct usage patterns and product themes. Traditional models often overlook the scene feature or treat it superficially, leading to potential information loss and failing to capture the complex interdependencies between scenes and items.

To tackle these issues, the authors propose a Dual Sequence Prediction network (DSPnet), a novel approach that simultaneously predicts scene and item sequences while capturing their inter-dependencies. DSPnet comprises two parallel networks for scene and item predictions and a sequence feature enhancement module to integrate these dependencies. To improve the model's robustness against the randomness and noise inherent in sequence data, the authors introduce a Conditional Contrastive Regularization (CCR) loss, which helps maintain the invariance of similar historical sequences during training.

Theoretical analysis indicates that DSPnet can effectively learn the joint relationships between scene and item sequences, thereby enhancing the accuracy of future behavior prediction. The effectiveness of DSPnet is validated through extensive experiments on a public benchmark dataset and two proprietary industrial datasets. The authors plan to make the source code and datasets publicly available to facilitate further research.

**Strengths:**

The proposed Dual Sequence Prediction network (DSPnet) method exhibits several strengths:

1. **Innovative Modeling of Inter-Dependencies**: DSPnet captures the inter-dependencies between scene and item sequences, addressing a critical gap in existing sequential behavior modeling methods. By using two parallel networks and a sequence feature enhancement module, it effectively integrates the dynamics of both scenes and items, leading to more accurate and comprehensive behavior predictions.

2. **Theoretical Robustness**: The theoretical analysis demonstrates that training DSPnet is equivalent to maximizing the joint log-likelihood of both scene and item sequences. This ensures that the model can effectively learn and represent the relationships between scenes and items, enhancing its predictive power.

3. **Conditional Contrastive Regularization (CCR)**: The introduction of CCR helps in capturing the invariance of similar historical sequences, which is crucial for dealing with the randomness and noise in user behavior data. CCR uses learned conditional weights to promote similarity among sequences, thereby improving the robustness and generalizability of the model, especially in scenarios with skewed user behaviors.

4. **Rich and Diverse Datasets**: The authors have collected 37 days of sequential user behavior data from their e-commerce app, constructing two industrial datasets. These datasets, along with a public benchmark, provide a rich and diverse set of data for validating the effectiveness of DSPnet. The datasets contain chronological purchase behaviors on nearly thirty million items, addressing the research data gap in this field.

5. **Empirical Validation**: Extensive experiments on three datasets—one public benchmark and two industrial datasets—demonstrate the superior performance of DSPnet compared to state-of-the-art baselines. The results highlight the importance of incorporating scene information in sequential behavior modeling and showcase the practical benefits of the proposed method.

These strengths collectively position DSPnet as a significant advancement in the field of sequential user behavior modeling, offering improved accuracy and robustness in predicting future user actions.

**Weaknesses:**

- The paper does not explicitly discuss the potential for integrating Conditional Contrastive Regularization (CCR) with other recommendation models.

**Questions:**

- Can Conditional Contrastive Regularization (CCR) be integrated with other recommendation models and be effective?

- Please provide an analysis of the model's complexity and efficiency and how to tune the hyperparameters for the training loss?

---

> ### Author Response · Authors · 2024-11-24
> **Response to Reviewer FSwH**
>
> ==============================================================================
> **W1 and Q1:**
>
> Yes. CCR can be integrated with other recommendation models. We have integrated it with the most competitive SOTA SceneContraRec, and the results are provided in the following table. The results clearly indicate that incorporating CCR loss enhances the performance of these sequential recommendation models, demonstrating its effectiveness.
>
> **Table: Incorporating CCR with other baseline models.**
> | Method             | Outbrain R@5 | Outbrain N@5 | Outbrain R@10 | Outbrain N@10 | AllScenePay-1m R@5 | AllScenePay-1m N@5 | AllScenePay-1m R@10 | AllScenePay-1m N@10 |
> |--------------------|--------------|--------------|---------------|---------------|--------------------|--------------------|---------------------|---------------------|
> | SceneContraRec     | 0.4979       | 0.4027       | 0.5448        | 0.4182        | 0.0762             | 0.0544             | 0.1045              | 0.0635              |
> | SceneContraRec+CCR | 0.4966       | 0.4088       | 0.5587        | 0.4250        | 0.0787             | 0.0571             | 0.1033              | 0.0651              |
>
> ==============================================================================
> **Q2:**
> We analyze the complexity of different models via two components: feature encoding and behavior prediction, both of which are commonly present in sequential behavior prediction models.
>
> The compared methods all use the powerful and popular transformer-based encoder.
> Let $B$ be the batch size, $L$ be the number of transformer layers, $|\mathcal{T}|$ be the sequence length of samples, $H$ be the head number, and $d$ be the dimension of each head. **The time complexity of the transformer-based encoder is $O(B \times L \times H \times |\mathcal{T}|^2 \times d)$, which is the same for all compared methods. The main difference in complexity lies in behavior prediction.** First, we denote $K^{v}$ as the number of candidate items (including positive and negative ones) for prediction. The complexity comparison is listed in the following table.
>
> **Table: Time complexity of different models on behavior prediction part.**
> |                |                                 |                                                                                                       |
> |----------------|---------------------------------|-------------------------------------------------------------------------------------------------------|
> | Method         | Behavior Prediction             | Remark                                                                                                |
> | Bert4Rec       | $O(B*\rho*\mathcal{T}*K^{v})$ | $\rho$ is the ratio of sequence tokens for Cloze task                                                 |
> | CARCA          |  $O(B*K^{v} + B *H' *(K^{v} *N') * d )$ | It involves cross attention between user-side features and candidate item. $H'$ is number of heads in cross attention and $N'$ is the number of user-side features |
> | SceneCTC       | $O(B*K^{v})$                    | It has no contrastive loss                                                                            |
> | MSDP           | $O(B * K^{v} + B^{2}*d)$              | It involves contrastive loss of input sequence                                                        |
> | ContraRec      | $O(B *K^{v} + B^{2} *d)$              | It involves contrastive loss of input sequence                                                        |
> | SceneContraRec | $O(B *K^{v} + B^2 *d)$              | It involves contrastive loss of input sequence                                                        |
> | DSPnet         | $O(B *K^{v} + 2 \cdot B^2 *d)$            | It involves CCR loss of two input sequences                                                           |
>
> In this table, $K^{v}$ generally reaches the magnitude of millions in industrial setting, while $|\mathcal{T}|$ is set it to $100$ in our large-scale datasets. **Consequently, the complexity of DSPnet is considerably lower than that of CARCA and Bert4Rec and does not significantly increase over SOTA methods such as SceneContraRec.** This makes DSPnet well-suited for usage in large-scale industrial datasets. We have successfully deployed it in our system using 16 A100 GPUs.
>
> About hyper-parameter tuning, we make grid search of hyper-parameters on a small scale dataset AllScenePay-1m, and directly use the configurations for AllScenePay-10m and more large-scale datasets. The sensitivity of hyper-parameters are given in Appendix-Figure 6 of our original paper.

---

> ### Author Response · Authors · 2024-11-27
> **Response to reviewer FSwH**
>
> Dear Reviewer FSwH, we sincerely appreciate your valuable suggestions and comments. As the comment phase is drawing to a close, we would like to know if all of your concerns have been adequately addressed. If you have any further questions or require additional clarification on any aspect of our work, we would be glad to provide thorough responses.

---

### Author Response · Authors · 2024-11-24
**Summary of Key Questions to Reviewers and ACs**

First, we would like to thank the reviewers for their valuable comments and constructive suggestions on the previous version of our submission. According to the comments, we have made the following changes in the revised version(changes are  marked with blue color in the new submitted pdf):

1. **Considering there is a misunderstanding of our scene feature for reviewers, we have clearly defined our scene features, setting our work apart from studies focused on multi-type behaviors.** Integrating our scene features into sequential behavior modeling presents an important problem derived from real-world industrial applications, which remains largely unexplored due to previously inaccessible data. We plan to make our extensive industrial datasets publicly available to the research community.

2. **We have highlighted the contributions of our work.** The proposed DSPnet is specifically designed to model sequential user behaviors with scenes. Its two main components: dual sequence learning with sequence feature enhancement and CCR loss, are both designed to address practical challenges in modeling sequential user behaviors with scenes.

3. **We have included a complexity analysis of various models.** This analysis demonstrates that our DSPnet either has significantly lower complexity or does not substantially increase compared to recent state-of-the-art (SOTA) methods. Additionally, DSPnet has been successfully deployed in our system using 16 A100 GPUs.

4. **We have conducted additional experiments including scene prediction and comparisons with more baselines to demonstrate the superiority of our method. Furthermore, we have analyzed and included more recent works in the reference.**

5. **We illustrate the problem difference, technical difference and result difference between our work and multi-behavior works.** We demonstrate the misalignment issue between scene and item in data, and why our dual sequence-level correspondence encoder outperforms the one-to-one correspondence encoding in multi-behavior works by additional experiments and comparisons.

We summarize the comments into the following sub-problems and answer each one separately.

Thank you for your kind consideration!


==============================================================================
**The used references in following response are:**
**Reference:**
1. Session-based Recommendation with Recurrent Neural Networks, ICLR'2016
2. BERT4Rec: Sequential Recommendation with Bidirectional Encoder Representations from Transformer, CIKM'2019
3. Sequential Recommendation with Multiple Contrast Signals, TOIS'2023
4. CARCA: Context and Attribute-Aware Next-Item Recommendation via Cross-Attention, RecSys'2022
5. Enhancing Sequential Recommendation via LLM-based Semantic Embedding Learning, WWW'2024
6. CALRec: Contrastive Alignment of Generative LLMs for Sequential Recommendation, RecSys'2024
7. Harnessing Large Language Models for Text-Rich Sequential Recommendation, WWW'2024
8. LLaRA: Large Language-Recommendation Assistant, ACM SIGIR'2024
9. Plug-In Diffusion Model for Sequential Recommendation, AAAI'2024
10. Generate what you prefer: Reshaping sequential recommendation via guided diffusion, NeurIPS'2024
11. Conditional denoising diffusion for sequential recommendation, PAKDD'2024
12. Linear Recurrent Units for Sequential Recommendation, WSDM'2024
13. Mamba4rec: Towards efficient sequential recommendation with selective state space models
14. Modeling Task Relationships in Multi-task Learning with Multi-gate Mixture-of-Experts, KDD'2018
15. Progressive Layered Extraction (PLE): A Novel Multi-Task Learning (MTL) Model for Personalized Recommendations, RecSys'2020
16. Scene-adaptive knowledge distillation for sequential recommendation via differentiable architecture search
17. SceneRec: Scene-Based Graph Neural Networks for Recommender Systems
18. LARR: Large Language Model Aided Real-time Scene Recommendation with Semantic Understanding, RecSys'2024
19. Scene-wise Adaptive Network for Dynamic Cold-start Scenes Optimization in CTR Prediction, RecSys'2024

---

---

### Comment · Area_Chair_piHv · 2024-11-30
**The deadline for Author/Reviewer discussion period is in three days!**

Dear Reviewers,

Thanks again for providing your constructive comments and suggestions. The deadline for the Author/Reviewer discussion period is in three days (December 2). Please make sure to read the authors' responses and follow up with them if you have any additional questions or feedback.

Best,

AC

---

### Meta-Review · Area_Chair_piHv · 2024-12-19

**Metareview:**

The paper presents a Dual Sequence Prediction network (DSPnet), aiming to capture the interplay between scene and item sequences for future behavior prediction.

**Strengths**
- It is interesting to study the scene feature, which provides critical contextual information to understand user behavior.
- The Conditional Contrastive Regularization (CCR) loss provides an effective way to capture the invariance of similar historical sequences.
- The effectiveness of the proposed approach is justified through both theoretical analysis and empirical evaluation.

**Weaknesses**
- While the paper provides examples on the scene feature, a more precise definition of a "scene" in the context of user behavior modeling is missing.
- The absence of a clear scene definition creates confusion regarding its distinction from multi-behavior modeling approaches, as noted by several reviewers.

Without a clear scene definition and more thorough distinction from existing methods in multi-behavior modeling, the overall contribution of the paper is not clearly established to meet the acceptance criteria.

**Additional Comments On Reviewer Discussion:**

During the rebuttal period, multiple reviewers engaged in discussions with the authors. While the authors’ responses addressed some of the reviewers’ concerns, key issues summarized in the weaknesses of the meta-review remain insufficiently addressed. The authors are encouraged to carefully consider these important suggestions to further improve their work for a future submission.

---

### Decision · Program_Chairs · 2025-01-22

Reject